# Multiway clustering via tensor block models

**Miaoyan Wang**
University of Wisconsin – Madison
miaoyan.wang@wisc.edu

**Yuchen Zeng**
University of Wisconsin – Madison
yzeng58@wisc.edu

## Abstract

We consider the problem of identifying multiway block structure from a large noisy tensor. Such problems arise frequently in applications such as genomics, recommendation system, topic modeling, and sensor network localization. We propose a tensor block model, develop a unified least-square estimation, and obtain the theoretical accuracy guarantees for multiway clustering. The statistical convergence of the estimator is established, and we show that the associated clustering procedure achieves partition consistency. A sparse regularization is further developed for identifying important blocks with elevated means. The proposal handles a broad range of data types, including binary, continuous, and hybrid observations. Through simulation and application to two real datasets, we demonstrate the outperformance of our approach over previous methods.

## 1 Introduction

Higher-order tensors have recently attracted increased attention in data-intensive fields such as neuroscience [1], social networks [2], computer vision [3], and genomics [4, 5]. In many applications, the data tensors are often expected to have underlying block structure. One example is multi-tissue expression data [4], in which genome-wide expression profiles are collected from different tissues in a number of individuals. There may be groups of genes similarly expressed in subsets of tissues and individuals; mathematically, this implies an underlying three-way block structure in the data tensor. In a different context, block structure may emerge in a binary-valued tensor. Examples include multilayer network data [2], with the nodes representing the individuals and the layers representing the multiple types of relations. Here a planted block represents a community of individuals that are highly connected within a class of relationships.

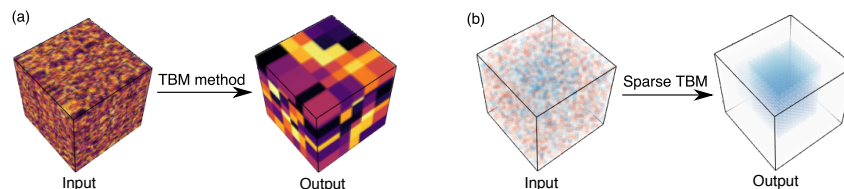

Figure 1: Examples of tensor block model (TBM). (a) Our TBM method is used for multiway clustering and for revealing the underlying checkerbox structure in a noisy tensor. (b) The sparse TBM method is used for detecting sub-tensors of elevated means.

This paper presents a new method and the associated theory for tensors with block structure. We develop a unified least-square estimation procedure for identifying multiway block structure. The proposal applies to a broad range of data types, including binary, continuous, and hybrid observations. We establish a high-probability error bound for the resulting estimator, and show that the procedure enjoys consistency guarantees on the block structure recovery as the dimension of the data tensor grows. Furthermore, we develop a sparse extension of the tensor block model for block selections.

Figure 1 shows two immediate examples of our method. When the data tensor possesses a checkerbox pattern modulo some unknown reordering of entries, our method amounts to multiway clustering that simultaneously clusters each mode of the tensor (Figure 1a). When the data tensor has no full checkerbox structure but contains a small numbers of sub-tensors of elevated means, we develop a sparse version of our method to detect these sub-tensors of interest (Figure 1b).

**Related work.** Our work is closely related to, but also clearly distinctive from, the low-rank tensor decomposition. A number of methods have been developed for low-rank tensor estimation, including CANDECOMP/PARAFAC (CP) decomposition [6] and Tucker decomposition [7]. The CP model decomposes a tensor into a sum of rank-1 tensors, whereas Tucker model decomposes a tensor into a core tensor multiplied by orthogonal matrices in each mode. In this paper we investigate an alternative block structure assumption, which has yet to be studied for higher-order tensors. Note that a block structure automatically implies low-rankness. However, as we will show in Section 4, a direct application of low rank estimation to the current setting will result in an inferior estimator. Therefore, a full exploitation of the block structure is necessary; this is the focus of the current paper.

Our work is also connected to biclustering [8] and its higher-order extensions [9, 10]. Existing multiway clustering methods [9, 10, 5, 11] typically take a two-step procedure, by first estimating a low-dimension representation of the data tensor and then applying clustering algorithms to the tensor factors. In contrast, our tensor block model takes a single shot to perform estimation and clustering simultaneously. This approach achieves a higher accuracy and an improved interpretability. Moreover, earlier solutions to multiway clustering [12, 9] focus on the algorithm effectiveness, leaving the statistical optimality of the estimators unaddressed. Very recently, Chi et al [13] provides an attempt to study the statistical properties of the tensor block model. We will show that our estimator obtains a faster convergence rate than theirs, and the power is further boosted with a sparse regularity.

## 2    Preliminaries

We begin by reviewing a few basic factors about tensors [14]. We use $\mathcal{Y} = [\![y_{i_1,\ldots,i_K}]\!] \in \mathbb{R}^{d_1 \times \cdots \times d_K}$ to denote an order-$K$ $(d_1,\ldots,d_K)$-dimensional tensor. The multilinear multiplication of a tensor $\mathcal{Y} \in \mathbb{R}^{d_1 \times \cdots \times d_K}$ by matrices $\boldsymbol{M}_k = [\![m_{i_k,j_k}^{(k)}]\!] \in \mathbb{R}^{s_k \times d_k}$ is defined as

$$\mathcal{Y} \times_1 \boldsymbol{M}_1 \ldots \times_K \boldsymbol{M}_K = [\![ \sum_{i_1,\ldots,i_K} y_{i_1,\ldots,i_K} m_{i_1,j_1}^{(1)} \ldots m_{i_K,j_K}^{(K)} ]\!],$$

which results in an order-$K$ tensor $(s_1,\ldots,s_K)$-dimensional tensor. For any two tensors $\mathcal{Y} = [\![y_{i_1,\ldots,i_K}]\!]$, $\mathcal{Y}' = [\![y'_{i_1,\ldots,i_K}]\!]$ of identical order and dimensions, their inner product is defined as $\langle \mathcal{Y}, \mathcal{Y}' \rangle = \sum_{i_1,\ldots,i_K} y_{i_1,\ldots,i_K} y'_{i_1,\ldots,i_K}$. The Frobenius norm of tensor $\mathcal{Y}$ is defined as $\|\mathcal{Y}\|_F = \langle \mathcal{Y}, \mathcal{Y} \rangle^{1/2}$; it is the Euclidean norm of $\mathcal{Y}$ regarded as an $\prod_k d_k$-dimensional vector. The maximum norm of tensor $\mathcal{Y}$ is defined as $\|\mathcal{Y}\|_{\max} = \max_{i_1,\ldots,i_K} |y_{i_1,\ldots,i_K}|$. An order-($K$-1) slice of $\mathcal{Y}$ is a sub-tensor of $\mathcal{Y}$ obtained by holding the index in one mode fixed while letting other indices vary.

A clustering of $d$ objects is a partition of the index set $[d] := \{1, 2, \ldots, d\}$ into $R$ disjoint non-empty subsets. We refer to the number of clusters, $R$, as the clustering size. Equivalently, the clustering (or partition) can be represented using the "membership matrix". A membership matrix $\boldsymbol{M} \in \mathbb{R}^{R \times d}$ is an incidence matrix whose $(i,j)$-entry is 1 if and only if the element $j$ belongs to the cluster $i$, and 0 otherwise. Throughout the paper, we will use the terms "clustering", "partition", and "membership matrix" exchangeably. For a higher-order tensor, the concept of index partition applies to each of the modes. A block is a sub-tensor induced by the index partitions along each of the $K$ modes. We use the term "cluster" to refer to the marginal partition on mode $k$, and reserve the term "block" for the multiway partition of the tensor.

## 3    Tensor block model

Let $\mathcal{Y} = [\![y_{i_1,\ldots,i_K}]\!] \in \mathbb{R}^{d_1 \times \cdots \times d_K}$ denote an order-$K$, $(d_1,\ldots,d_K)$-dimensional data tensor. The main assumption of tensor block model (TBM) is that the observed data tensor $\mathcal{Y}$ is a noisy realization of an underlying tensor that exhibits a checkerbox structure (see Figure 1a). Specifically, suppose that the $k$-th mode of the tensor consists of $R_k$ clusters. If the tensor entry $y_{i_1,\ldots,i_K}$ belongs to the block determined by the $r_k$th cluster in the mode $k$ for $r_k \in [R_k]$, then we assume that

$$y_{i_1,\ldots,i_K} = c_{r_1,\ldots,r_K} + \varepsilon_{i_1,\ldots,i_K}, \quad \text{for } (i_1,\ldots,i_K) \in [d_1] \times \cdots \times [d_K], \tag{1}$$

where $c_{r_1,\ldots,r_K}$ is the mean of the tensor block indexed by $(r_1,\ldots,r_K)$, and $\varepsilon_{i_1,\ldots,i_K}$'s are independent, mean-zero noise terms to be specified later. Our goal is to (i) find the clustering along each of the modes, and (ii) estimate the block means $\{c_{r_1,\ldots,r_K}\}$, such that a corresponding blockwise-constant checkerbox structure emerges in the data tensor.

The tensor block model (1) falls into a general class of non-overlapping, constant-mean clustering models [15], in that each tensor entry belongs to exactly one block with a common mean. The TBM can be equivalently expressed as a special tensor Tucker model,

$$\mathcal{Y} = \mathcal{C} \times_1 \boldsymbol{M}_1 \times_2 \cdots \times_K \boldsymbol{M}_K + \mathcal{E}, \tag{2}$$

where $\mathcal{C} = [\![c_{r_1,\ldots,r_K}]\!] \in \mathbb{R}^{R_1 \times \cdots \times R_K}$ is a core tensor consisting of block means, $\boldsymbol{M}_k \in \{0,1\}^{d_k \times R_k}$ is a membership matrix indicating the block allocations along mode $k$ for $k \in [K]$, and $\mathcal{E} = [\![\varepsilon_{i_1,\ldots,i_K}]\!]$ is the noise tensor. We view the TBM (2) as a super-sparse Tucker model, in the sense that the each column of $\boldsymbol{M}_k$ consists of one copy of 1's and massive 0's.

We make a general assumption on the noise tensor $\mathcal{E}$. The noise terms $\varepsilon_{i_1,\ldots,i_K}$'s are assumed to be independent, mean-zero $\sigma$-subgaussian, where $\sigma > 0$ is the subgaussianity parameter. Precisely,

$$\mathbb{E}e^{\lambda \varepsilon_{i_1,\ldots,i_K}} \leq e^{\lambda^2 \sigma^2 / 2}, \quad \text{for all } (i_1,\ldots,i_K) \in [d_1] \times \cdots \times [d_K] \text{ and all } \lambda \in \mathbb{R}. \tag{3}$$

Th assumption (3) incorporates common situations such as Gaussian noise, Bernoulli noise, and noise with bounded support. In particular, we consider two important examples of the TBM:

**Example 1 (Gaussian tensor block model)** *Let $\mathcal{Y}$ be a continuous-valued tensor. The Gaussian tensor block model (GTBM) $y_{i_1,\ldots,i_K} \sim_{i.i.d.} N(c_{r_1,\ldots,r_K}, \sigma^2)$ is a special case of model (1), with the subgaussianity parameter $\sigma$ equal to the error variance. The GTBM serves as the foundation for many tensor clustering algorithms [12, 4, 13].*

**Example 2 (Stochastic tensor block model)** *Let $\mathcal{Y}$ be a binary-valued tensor. The stochastic tensor block model (STBM) $y_{i_1,\ldots,i_K} \sim_{i.i.d.} \text{Bernoulli}(c_{r_1,\ldots,r_K})$ is a special case of model (1), with the subgaussianity parameter $\sigma$ equal to $\frac{1}{4}$. The STBM can be viewed as an extension, to higher-order tensors, of the popular stochastic block model [16, 17] for matrix-based network analysis. In the filed of community detection, multi-layer stochastic model has also been developed for multi-relational network data analysis [18, 19].*

More generally, our model also applied to hybrid error distributions, in which different types of distribution are allowed for different portions of the tensor. This scenario may happen, for example, when the data tensor $\mathcal{Y}$ represents concatenated measurements from multiple data sources.

Before we discuss the estimation, we present the identifiability of the TBM.

**Assumption 1 (Irreducible core)** *The core tensor $\mathcal{C}$ is called irreducible if it cannot be written as a block tensor with the number of mode-$k$ clusters smaller than $R_k$, for any $k \in [K]$.*

In the matrix case $(K = 2)$, the irreducibility is equivalent to saying that $\mathcal{C}$ has no two identical rows and no two identical columns. In the higher-order case, the assumption requires that none of order-$(K\text{-}1)$ slices of $\mathcal{C}$ are identical. Note that irreducibility is a weaker assumption than full-rankness.

**Proposition 1 (Identifiability)** *Consider a Gaussian or Bernoulli TBM (1). Under Assumption 1, the factor matrices $\boldsymbol{M}_k$'s are identifiable up to permutations of cluster labels.*

The identifiability property for the TBM outperforms that for the classical factor model [20, 21]. In the Tucker [22, 14] and many other factor analyses [20, 21], the factors are identifiable only up to orthogonal rotations. Those models recover only the (column) space spanned by $\boldsymbol{M}_k$, but not the individual factors. In contrast, our model does not suffer from rotational invariance, and as we show in Section 4, every individual factor is consistently estimated in high dimensions. This brings a benefit to the interpretation of factors in the tensor block model.

We propose a least-square approach for estimating the TBM. Let $\Theta = \mathcal{C} \times_1 \boldsymbol{M}_1 \times_2 \cdots \times_K \boldsymbol{M}_K$ denote the mean signal tensor with block structure. The mean tensor is assumed to belong to the following parameter space

$$\mathcal{P}_{R_1,\ldots,R_K} = \big\{ \Theta \in \mathbb{R}^{d_1 \times \cdots \times d_K} : \Theta = \mathcal{C} \times_1 \boldsymbol{M}_1 \times_2 \cdots \times_K \boldsymbol{M}_K, \text{ with some membership matrices}$$
$$\boldsymbol{M}_k\text{'s and a core tensor } \mathcal{C} \in \mathbb{R}^{R_1 \times \cdots \times R_K} \big\}.$$

In the following theoretical analysis, we assume the clustering size $\boldsymbol{R} = (R_1, \ldots, R_K)$ is known and simply write $\mathcal{P}$ for short. The adaptation of unknown $\boldsymbol{R}$ will be addressed in Section 5.2. The least-square estimator for the TBM (1) is

$$\hat{\Theta} = \arg\min_{\Theta \in \mathcal{P}} \left\{ -2\langle \mathcal{Y}, \Theta \rangle + \|\Theta\|_F^2 \right\}. \tag{4}$$

The objective is equal (ignoring constants) to the sum of squares $\|\mathcal{Y} - \Theta\|_F^2$ and hence the name of our estimator.

## 4 Statistical convergence

In this section, we establish the convergence rate of the least-squares estimator (4) for two measurements. The first measurement is mean squared error (MSE):

$$\text{MSE}(\Theta_{\text{true}}, \hat{\Theta}) = \frac{1}{\prod_k d_k} \|\Theta_{\text{true}} - \hat{\Theta}\|_F^2,$$

where $\Theta_{\text{true}}, \hat{\Theta} \in \mathcal{P}$ are the true and estimated mean tensors, respectively. While the loss function corresponds to the likelihood for the Gaussian tensor model, the same assertion does not hold for other types of distribution such as stochastic tensor block model. We will show that, with very high probability, a simple least-square estimator achieves a fast convergence rate in a general class of block tensor models.

**Theorem 1 (Convergence rate of MSE)** *Let $\hat{\Theta}$ be the least-square estimator of $\Theta_{true}$ under model* (1). *There exists two constants $C_1, C_2 > 0$, such that,*

$$MSE(\Theta_{true}, \hat{\Theta}) \leq \frac{C_1 \sigma^2}{\prod_k d_k} \left( \prod_k R_k + \sum_k d_k \log R_k \right) \tag{5}$$

*holds with probability at least $1 - \exp(-C_2(\prod_k R_k + \sum_k d_k \log R_k))$ uniformly over $\Theta_{true} \in \mathcal{P}$.*

The convergence rate of MSE in (5) consists of two parts. The first part $\prod_k R_k$ is the number of parameters in the core tensor $\mathcal{C}$, while the second part $\sum_k d_k \log R_k$ reflects the the complexity for estimating $\boldsymbol{M}_k$'s. It is the price that one has to pay for not knowing the locations of the blocks.

We compare our bound with existing literature. The Tucker tensor decomposition has a minimax convergence rate proportional to $\sum_k d_k R'_k$ [22], where $R'_k$ is the multilinear rank in the mode $k$. Applying Tucker decomposition to the TBM yields $\sum_k d_k R_k$, because the mode-$k$ rank is bounded by the number of mode-$k$ clusters. Now, as both the dimension $d_{\min} = \min_k d_k$ and clustering size $R_{\min} = \min_k R_k$ tend to infinity, we have $\prod_k R_k + \sum_k d_k \log R_k \ll \sum_k d_k R_k$. Therefore, by fully exploiting the block structure, we obtain a better convergence rate than previously possible.

Recently, [13] proposed a convex relaxation for estimating the TBM. In the special case when the tensor dimensions are equal at every mode $d_1 = \ldots = d_K = d$, their estimator has a convergence rate of order $\mathcal{O}(d^{-1})$ for all $K \geq 2$. As we see from (5), our estimate obtains a much better convergence rate $\mathcal{O}(d^{-(K-1)})$, which is especially favorable as the order increases.

The bound (5) generalizes the previous results on structured matrix estimation in network analysis [23, 16]. Earlier work [16] suggests the following heuristics on the sample complexity for the matrix case:

$$\frac{\text{(number of parameters)} + \log \text{(complexity of models)}}{\text{number of samples}}. \tag{6}$$

Our result supports this important principle for general $K \geq 2$. Note that, in the TBM, the sample size is the total number of entries $\prod_k d_k$, the number of parameters is $\prod_k R_k$, and the combinatoric complexity for estimating block structure is of order $\prod_k R_k^{d_k}$.

Next we study the consistency of partition. To define the misclassification rate (MCR), we need to introduce some additional notation. Let $\boldsymbol{M}_k = [\![m_{i,r}^{(k)}]\!], \hat{\boldsymbol{M}}_k = [\![\hat{m}_{i,r'}^{(k)}]\!]$ be two mode-$k$ membership matrices, and $\boldsymbol{D}^{(k)} = [\![D_{r,r'}^{(k)}]\!]$ be the mode-$k$ confusion matrix with element $D_{r,r'}^{(k)} = \frac{1}{d_k} \sum_{i=1}^{d_k} \mathbb{1}\{m_{i,r}^{(k)} = \hat{m}_{i,r'}^{(k)} = 1\}$, where $r, r' \in [R_k]$. Note that the row/column sum

of $\boldsymbol{D}^{(k)}$ represents the nodes proportion in each cluster defined by $\boldsymbol{M}_k$ or $\hat{\boldsymbol{M}}_k$. We restrict ourselves to non-degenerating clusterings; that is, the row/column sums of $\boldsymbol{D}^{(k)}$ are lower bounded by a constant $\tau > 0$. With a little abuse of notation, we still use $\mathcal{P} = \mathcal{P}(\tau)$ to denote the parameter space with the non-degenerating assumption. The least-square estimator (4) should also be interpreted with this constraint imposed.

We define the mode-$k$ misclassification rate (MCR) as

$$\text{MCR}(\boldsymbol{M}_k, \hat{\boldsymbol{M}}_k) = \max_{r \in [R_k], a \neq a' \in [R_k]} \min \left\{ D_{a,r}^{(k)}, \; D_{a',r}^{(k)} \right\}.$$

In other words, MCR is the element-wise maximum of the confusion matrix after removing the largest entry from each column. Under the non-degenerating assumption, MCR $= 0$ if and only if the confusion matrix $\boldsymbol{D}^{(k)}$ is a permutation of a diagonal matrix; that is, the estimated partition matches with the true partition, up to permutations of cluster labels.

**Theorem 2 (Convergence rate of MCR)** *Consider a tensor block model* (2) *with sub-Gaussian parameter $\sigma$. Define the minimal gap between the blocks $\delta_{\min} = \frac{1}{\|\mathcal{C}\|_{\max}} \min_k \delta^{(k)}$, where $\delta^{(k)} = \min_{r_k \neq r_k'} \max_{r_1,\ldots,r_{k-1},r_{k+1},\ldots,r_K} (c_{r_1,\ldots,r_k,\ldots,r_K} - c_{r_1,\ldots,r_k',\ldots,r_K})^2$. Let $\boldsymbol{M}_{k,true}$ be the true mode-$k$ membership, $\hat{\boldsymbol{M}}_k$ be the estimator from* (4). *Then, for any $\varepsilon \in [0,1]$,*

$$\mathbb{P}(MCR(\hat{\boldsymbol{M}}_k, \boldsymbol{M}_{k,true}) \geq \varepsilon) \leq 2^{1 + \sum_k d_k} \exp\left( -\frac{C\varepsilon^2 \delta_{\min}^2 \tau^{3K-2} \prod_{k=1}^K d_k}{\sigma^2} \right),$$

*where $C > 0$ is a positive constant, and $\tau > 0$ the lower bound of cluster proportions.*

The above theorem shows that our estimator consistently recovers the block structure as the dimension of the data tensor grows. The block-mean gap $\delta_{\min}$ serves the role of the eigen-separation as in the classical tensor Tucker decomposition [22]. Table 1 summarizes the comparison of various tensor methods in the special case when $d_1 = \cdots = d_K = d$ and $R_1 = \cdots = R_K = R$.

| Method | Recovery error ($\|\hat{\Theta} - \Theta_{\text{true}}\|_F^2 / \sigma^2$) | Clustering error (MCR) | Block detection (see Section 6) |
|---|---|---|---|
| Tucker [22] | $dR$ | - | No |
| CoCo [13] | $d^{K-1}$ | - | No |
| TBM (this paper) | $d \log R$ | $\frac{\sigma}{\delta_{\min} \tau^{(3K-2)/2}} d^{-(K-1)/2}$ | Yes |

Table 1: Comparison of various tensor decomposition methods. For ease of presentation, we summarize only the leading terms in dimension.

## 5 Numerical implementation

### 5.1 Alternating optimization

We introduce an alternating optimization for solving (4). Estimating $\Theta$ consists of finding both the core tensor $\mathcal{C}$ and the membership matrices $\boldsymbol{M}_k$'s. The optimization (4) can be written as

$$(\hat{\mathcal{C}}, \{\hat{\boldsymbol{M}}_k\}) = \arg\min_{\mathcal{C} \in \mathbb{R}^{R_1 \times \cdots \times R_K}, \text{ membership matrices } \boldsymbol{M}_k\text{'s}} f(\mathcal{C}, \{\boldsymbol{M}_k\}),$$

$$\text{where} \quad f(\mathcal{C}, \{\boldsymbol{M}_k\}) = \|\mathcal{Y} - \mathcal{C} \times_1 \boldsymbol{M}_1 \times_2 \ldots \times_K \boldsymbol{M}_K\|_F^2.$$

The decision variables consist of $K + 1$ blocks of variables, one for the core tensor $\mathcal{C}$ and $K$ for the membership matrices $\boldsymbol{M}_k$'s. We notice that, if any $K$ out of the $K + 1$ blocks of variables are known, then the last block of variables can be solved explicitly. This observation suggests that we can iteratively update one block of variables at a time while keeping others fixed. Specifically, given the collection of $\hat{\boldsymbol{M}}_k$'s, the core tensor estimate $\hat{\mathcal{C}} = \arg\min_\mathcal{C} f(\mathcal{C}, \{\hat{\boldsymbol{M}}_k\})$ consists of the sample averages of each tensor block. Given the block mean $\hat{\mathcal{C}}$ and $K - 1$ membership matrices, the last membership matrix can be solved using a simple nearest neighbor search over only $R_k$ discrete points. The full procedure is described in Algorithm 1.

Algorithm 1 can be viewed as a higher-order extension of the ordinary (one-way) $k$-means algorithm. The core tensor $\mathcal{C}$ serves as the role of centroids. As each iteration reduces the value of the objective function, which is bounded below, convergence of the algorithm is guaranteed. The per-iteration

---
**Algorithm 1** Multiway clustering based on tensor block models
---
**Input:** Data tensor $\mathcal{Y} \in \mathbb{R}^{d_1 \times \cdots \times d_K}$, clustering size $\boldsymbol{R} = (R_1, \ldots, R_K)$.
**Output:** Block mean tensor $\hat{\mathcal{C}} \in \mathbb{R}^{R_1 \times \cdots \times R_K}$, and the membership matrices $\hat{M}_k$'s.
 1: Initialize the marginal clustering by performing independent $k$-means on each of the $K$ modes.
 2: **repeat**
 3:      Update the core tensor $\hat{\mathcal{C}} = [\![\hat{c}_{r_1,\ldots,r_K}]\!]$. Specifically, for each $(r_1, \ldots, r_K) \in [R_1] \times \cdots [R_K]$,

$$\hat{c}_{r_1,\ldots,r_K} = \frac{1}{n_{r_1,\ldots,r_K}} \sum_{\hat{M}_1^{-1}(r_1) \times \cdots \times \hat{M}_K^{-1}(r_K)} y_{i_1,\ldots,i_K}, \tag{7}$$

     where $M_k^{-1}(r_k)$ denotes the indices that belong to the $r_k$th cluster in the mode $k$, and $n_{r_1,\ldots,r_K} = \prod_k |\hat{M}_k^{-1}(r_k)|$ denotes the number of entries in the block indexed by $(r_1, \ldots, r_K)$.
 4:      **for** $k$ in $\{1, 2, ..., K\}$ **do**
 5:          Update the mode-$k$ membership matrix $\hat{M}_k$. Specifically, for each $a \in [d_k]$, assign the cluster label $\hat{M}_k(a) \in [R_k]$:

$$\hat{M}_k(a) = \underset{r \in [R_k]}{\arg\min} \sum_{\boldsymbol{I}_{-k}} \left( \hat{c}_{\hat{M}_1(i_1),\ldots,r,\ldots,\hat{M}_K(i_K)} - y_{i_1,\ldots,a,\ldots,i_K} \right)^2,$$

         where $\boldsymbol{I}_{-k} = (i_1, \ldots, i_{k-1}, i_{k+1}, \ldots, i_K)$ denotes the tensor coordinates except the $k$-th mode.
 6:      **end for**
 7: **until** Convergence
---

computational cost scales linearly with the sample size, $d = \prod_k d_k$, and this complexity matches the classical tensor methods [24, 25, 22]. We recognize that obtaining the global optimizer for such a non-convex optimization is typically difficult [26, 1]. Following the common practice in non-convex optimization [1], we run the algorithm multiple times, using random initializations with independent one-way $k$-means on each of the modes.

### 5.2 Tuning parameter selection

Algorithm 1 takes the number of clusters $\boldsymbol{R}$ as an input. In practice such information is often unknown and $\boldsymbol{R}$ needs to be estimated from the data $\mathcal{Y}$. We propose to select this tuning parameter using Bayesian information criterion (BIC),

$$\text{BIC}(\boldsymbol{R}) = \log \left( \|\mathcal{Y} - \hat{\Theta}\|_F^2 \right) + \frac{\sum_k \log d_k}{\prod_k d_k} p_e, \tag{8}$$

where $p_e$ is the effective number of parameters in the model. In our case we take $p_e = \prod_k R_k + \sum_k d_k \log R_k$, which is inspired from (6). We choose $\hat{\boldsymbol{R}}$ that minimizes $\text{BIC}(\boldsymbol{R})$ via grid search. Our choice of BIC aims to balance between the goodness-of-fit for the data and the degree of freedom in the population model. We test its empirical performance in Section 7.

## 6 Extension to sparse estimation

In some large-scale applications, not every block in a data tensor is of equal importance. For example, in the genome-wide expression data analysis, only a few entries represent the signals while the majority come from the background noise (see Figure 1b). While our estimator (4) is still able to handle this scenario by assigning small values to some of the $\hat{c}_{r_1,\ldots,r_K}$'s, the estimates may suffer from high variance. It is thus beneficial to introduce regularized estimation for better bias-variance trade-off and improved interpretability.

Here we illustrate a sparse version of TBM by imposing regularity on block means for localizing important blocks in the data tensor. This problem can be formulated as a variable selection on the block parameters. We propose the following regularized least-square estimation:

$$\hat{\Theta}^{\text{sparse}} = \underset{\Theta \in \mathcal{P}}{\arg\min} \left\{ \|\mathcal{Y} - \Theta\|_F^2 + \lambda \|\mathcal{C}\|_\rho \right\},$$

where $\mathcal{C} \in \mathbb{R}^{R_1 \times \cdots \times R_K}$ is the block-mean tensor, $\|\mathcal{C}\|_\rho$ is the penalty function with $\rho$ being an index for the tensor norm, and $\lambda$ is the penalty tuning parameter. Some widely used penalties include Lasso penalty ($\rho = 1$), sparse subset penalty ($\rho = 0$), ridge penalty ($\rho =$ Frobenius norm), elastic net (linear combination of $\rho = 1$ and $\rho =$ Frobenius norm), among many others.

For parsimony purpose, we only discuss the Lasso and sparse subset penalties; other penalizations can be derived similarly. Sparse estimation incurs slight changes to Algorithm 1. When updating the core tensor $\mathcal{C}$ in (7), we fit a penalized least square problem with respect to $\mathcal{C}$. The closed form for the entry-wise sparse estimate $\hat{c}_{r_1,\dots,r_K}^{\text{sparse}}$ is (see Lemma 2 in the Supplements):

$$
\hat{c}_{r_1,\dots,r_K}^{\text{sparse}} = \begin{cases} \hat{c}_{r_1,\dots,r_K}^{\text{ols}} \mathbb{1}\left\{ |\hat{c}_{r_1,\dots,r_K}^{\text{ols}}| \geq \sqrt{\dfrac{\lambda}{n_{r_1,\dots,r_K}}} \right\} & \text{if } \rho = 0, \\ \text{sign}(\hat{c}_{r_1,\dots,r_K}^{\text{ols}}) \left( |\hat{c}_{r_1,\dots,r_K}^{\text{ols}}| - \dfrac{\lambda}{2n_{r_1,\dots,r_K}} \right)_+ & \text{if } \rho = 1, \end{cases}
$$

where $a_+ = \max(a,0)$ and $\hat{c}_{r_1,\dots,r_K}^{\text{ols}}$ denotes the ordinary least-square estimate in (7). The choice of penalty $\rho$ often depends on the study goals and interpretations in specific applications. Given a penalty function, we select the tuning parameter $\lambda$ via BIC (8), where we modify $p_e$ into $p_e^{\text{sparse}} = \|\hat{\mathcal{C}}^{\text{sparse}}\|_0 + \sum_k d_k \log R_k$. Here $\|\cdot\|_0$ denotes the number of non-zero entries in the tensor. The empirical performance of this proposal will be evaluated in Section 7.

# 7 Experiments

In this section, we evaluate the empirical performance of our TBM method[1]. We consider both non-sparse and sparse tensors, and compare the recovery accuracy with other tensor-based methods. Unless otherwise stated, we generate Gaussian tensors under the block model (1). The block means are generated from i.i.d. Uniform[-3,3]. The entries in the noise tensor $\mathcal{E}$ are generated from i.i.d. $N(0, \sigma^2)$. In each simulation study, we report the summary statistics across $n_{\text{sim}} = 50$ replications.

## 7.1 Finite-sample performance

In the first experiment, we assess the empirical relationship between the root mean squared error (RMSE) and the dimension. We set $\sigma = 3$ and consider tensors of order 3 and order 4 (see Figure 2). In the case of order-3 tensors, we increase $d_1$ from 20 to 70, and for each choice of $d_1$, we set the other two dimensions $(d_2, d_3)$ such that $d_1 \log R_1 \approx d_2 \log R_2 \approx d_3 \log R_3$. Recall that our theoretical analysis suggests a convergence rate $\mathcal{O}(\sqrt{\log R_1 / d_2 d_3})$ for our estimator. Figure 2a plots the recovery error versus the rescaled sample size $N_1 = \sqrt{d_2 d_3 / \log R_1}$. We find that the RMSE decreases roughly at the rate of $1/N_1$. This is consistent with our theoretical result. It is observed that tensors with a higher number of blocks tend to yield higher recovery errors, as reflected by the upward shift of the curves as $\boldsymbol{R}$ increases. Indeed, a higher $\boldsymbol{R}$ means a higher intrinsic dimension of the problem, thus increasing the difficulty of the estimation. Similar behavior can be observed in the order-4 case from Figure 2b, where the rescaled sample size is $N_2 = \sqrt{d_2 d_3 d_4 / \log R_1}$.

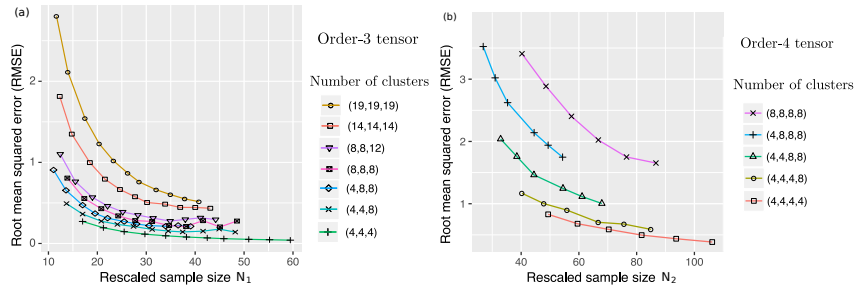

Figure 2: Estimation error for block tensors with Gaussian noise. Each curve corresponds to a fixed clustering size $\boldsymbol{R}$. (a) Average RMSE against rescaled sample size $N_1 = \sqrt{d_2 d_3 / \log R_1}$ for order-3 tensors. (b) Average RMSE against rescaled sample size $N_2 = \sqrt{d_2 d_3 d_4 / \log R_1}$ for order-4 tensors.

In the second experiment, we evaluate the selection performance of our BIC criterion (8). Supplementary Table S1 reports the selected numbers of clusters under various combinations of dimension $\boldsymbol{d}$,

[1]Our software is available at `https://cran.r-project.org/web/packages/tensorsparse`.

clustering size $\boldsymbol{R}$, and noise $\sigma$. We find that, for the case $\boldsymbol{d} = (40, 40, 40)$ and $\boldsymbol{R} = (4, 4, 4)$, the BIC selection is accurate in the low-to-moderate noise setting. In the high-noise setting with $\sigma = 12$, the selected number of clusters is slightly smaller than the true number, but the accuracy increases when either the dimension increases to $\boldsymbol{d} = (40, 40, 80)$ or the clustering size reduces to $\boldsymbol{R} = (2, 3, 4)$. Within a tensor, the selection seems to be easier for shorter modes with smaller number of clusters. This phenomenon is to be expected, since shorter mode has more effective samples for clustering.

## 7.2 Comparison with alternative methods

Next, we compare our TBM method with two popular low-rank tensor estimation methods: (i) CP decomposition and (ii) Tucker decomposition. Following the literature [13, 5, 9], we perform the clustering by applying the $k$-means to the resulting factors along each of the modes. We refer to such techniques as CP+$k$-means and Tucker+$k$-means.

We generate noisy block tensors with five clusters on each of the modes, and then assess both the estimation and clustering performance for each method. Note that TBM takes a single shot to perform estimation and clustering simultaneously, whereas CP and Tucker-based methods separate these two tasks in two steps. We use the RMSE to assess the estimation accuracy and use the clustering error rate (CER) to measure the clustering accuracy. The CER is calculated using the disagreements (i.e., one minus rand index) between the true and estimated block partitions in the three-way tensor. For fair comparison, we provide all methods the true number of clusters.

Figure 3a shows that TBM achieves the lowest estimation error among the three methods. The gain in accuracy is more pronounced as the noise grows. Neither CP nor Tucker recovers the signal tensor, although Tucker appears to result in a modest clustering performance (Figure 3b). One possible explanation is that the Tucker model imposes orthogonality to the factors, which make the subsequent $k$-means clustering easier than that for the CP factors. Figure 3b-c shows that the clustering error increases with noise but decreases with dimension. This agrees with our expectation, as in tensor data analysis, a larger dimension implies a larger sample size.

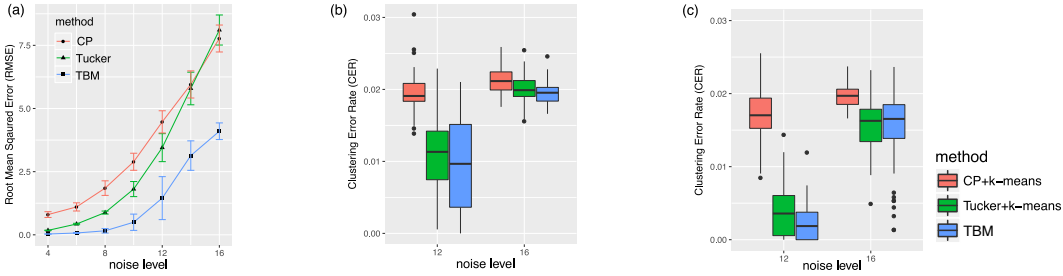

Figure 3: Performance comparison in terms of RMSE and CER. (a) Estimation error against noise for tensors of dimension $(40, 40, 40)$. (b) Clustering error against noise for tensors of dimension $(40, 40, 40)$. (c) Clustering error against noise for tensors of dimension $(40, 50, 60)$.

**Sparse case.** We then evaluate the performance when the signal tensor is sparse. The simulated model is the same as before, except that we generate block means from a mixture of zero mass and Uniform[-3,3], with probability $p$ (sparsity rate) and $1 - p$ respectively. We generate noisy tensors of dimension $\boldsymbol{d} = (40, 40, 40)$ with varying levels of sparsity and noise. We utilize $\ell 0$-penalized TBM and primarily focus on the selection accuracy. The performance is quantified via the the sparsity error rate, which is the proportion of entries that were incorrectly set to zero or incorrectly set to non-zero. We also report the proportion of true zero's that were correctly identified (correct zeros).

Table 2 reports the BIC-selected $\lambda$ averaged across 50 simulations. We see a substantial benefit obtained by penalization. The proposed $\lambda$ is able to guide the algorithm to correctly identify zero's, while maintaining good accuracy in identifying non-zero's. The resulting sparsity level is close to the ground truth. Supplementary Figure S1 shows the estimation error and sparsity error against $\sigma$ when $p = 0.8$. Again, the sparse TBM outperforms the other methods.

| Sparsity ($p$) | Noise ($\sigma$) | BIC-selected $\lambda$ | Estimated Sparsity Rate | Correct Zero Rate | Sparsity Error Rate |
|:---:|:---:|:---:|:---:|:---:|:---:|
| 0.5 | 4 | 136.0(37.5) | **0.55(0.04)** | **1.00(0.02)** | **0.06(0.03)** |
| 0.5 | 8 | 439.2(80.2) | **0.58(0.06)** | **0.94(0.08)** | 0.15(0.07) |
| 0.8 | 8 | 458.0(63.3) | **0.81(0.15)** | **0.87(0.16)** | **0.21(0.13)** |

Table 2: Sparse TBM for estimating tensors of dimension $\boldsymbol{d} = (40, 40, 40)$. The reported statistics are averaged across 50 simulations with standard deviation given in parentheses. Number in bold indicates the ground truth is within 2 standard deviations of the sample average.

## 7.3 Real data analysis

Lastly, we apply our method on two real datasets. The first dataset is a real-valued tensor, consisting of approximate 1 million expression values from 13 brain tissues, 193 individuals, and 362 genes [4]. We subtracted the overall mean expression from the data, and applied the $\ell 0$-penalized TBM to identify important blocks in the resulting tensor. The top blocks exhibit a clear tissues $\times$ genes specificity (Supplementary Table S2). In particular, the top over-expressed block is driven by tissues {*Substantia nigra, Spinal cord*} and genes {*GFAP, MBP*}, suggesting their elevated expression across individuals. In fact, *GFAP* encodes filament proteins for mature astrocytes and *MBP* encodes myelin sheath for oligodendrocytes, both of which play important roles in the central nervous system [27]. Our method also identifies blocks with extremely negative means (i.e. under-expressed blocks). The top under-expressed block is driven by tissues {*Cerebellum, Cerebellar Hemisphere*} and genes {*CDH9, GPR6, RXFP1, CRH, DLX5/6, NKX2-1, SLC17A8*}. The gene *DLX6* encodes proteins in the forebrain development [27], whereas cerebellum tissues are located in the hindbrain brain. The opposite spatial function is consistent with the observed under-expression pattern.

The second dataset we consider is the *Nations* data [2]. This is a $14 \times 14 \times 56$ binary tensor consisting of 56 political relationships of 14 countries between 1950 and 1965. We note that 78.9% of the entries are zero. Again, we applied the $\ell 0$-penalized TBM to identify important blocks in the data. We found that the 14 countries are naturally partitioned into 5 clusters, two representing neutral countries {*Brazil, Egypt, India, Israel, Netherlands*} and {*Burma, Indonesia, Jordan*}, one eastern bloc {*China, Cuba, Poland, USSA*}, and two western blocs, {*USA*} and {*UK*}. The relation types are partitioned into 7 clusters, among which the exports-related activities {*reltreaties, book translations, relbooktranslations, exports3, relexporsts*} and NGO-related activities {*relintergovorgs, relngo, intergovorgs3, ngoorgs3*} are two major clusters that involve the connection between neutral and western blocs. Other top blocks are described in the Supplement.

We compared the goodness-of-fit of various clustering methods on the *Brain expression* and *Nations* datasets. Because the code of CoCo method [13] is not yet available, we excluded it from our numerical comparison (See Section 4 for the theoretical comparison with CoCo). The Table 3 summarizes the proportion of variance explained by each clustering method:

| Dataset | TBM | TBM-sparse | CP+$k$-means | Tucker+$k$-means | CoTeC [12] |
|:---:|:---:|:---:|:---:|:---:|:---:|
| Brain expression | 0.856 | 0.855 | 0.576 | 0.434 | 0.849 |
| Nations | 0.439 | 0.433 | 0.324 | 0.253 | 0.419 |

Table 3: Comparison of goodness-of-fit in the *Brain* expression and *Nations* datasets.

Our method (TBM) achieves the highest variance proportion, suggesting that the entries within the same cluster are close (i.e., a good clustering). As expected, the sparse TBM results in a slightly lower proportion, because it has a lower model complexity at the cost of small bias. It is remarkable that the sparse TBM still achieves a higher goodness-of-fit than others. The improved interpretability with little loss of accuracy makes the sparse TBM appealing in applications.

## 8 Conclusion

We have developed a statistical setting for studying the tensor block model. Under suitable assumptions, the least-square estimator achieves a convergence rate $\mathcal{O}(\sum_k d_k \log R_k)$ which is faster than previously possible. Our TBM method applies to a broad range of data distributions and can handle both sparse and dense data tensor. We demonstrate the benefit of sparse regularity in power of detection. In specific applications, prior knowledge may suggest other regularities for parameters. For example, in the multi-layer network analysis, sometimes it may be reasonable to impose symmetry on the parameters along certain modes. In some other applications, non-negativity of parameter values may be enforced. We leave these directions for future study.

## Acknowledgements

This research was supported by NSF grant DMS-1915978 and the University of Wisconsin-Madison, Office of the Vice Chancellor for Research and Graduate Education with funding from the Wisconsin Alumni Research Foundation.

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
