[Supplementary Material · TBM_supp.pdf]

# Supplements for "Multiway clustering via tensor block models"

## A Proofs

### A.1 Stochastic tensor block model

The following property shows that Bernoulli distribution belongs to the sub-Gaussian family with a subgaussianity parameter $\sigma$ equal to $1/4$.

**Property 1.** *Suppose $x \sim Bernoulli(p)$, then $x \sim sub\text{-}Gaussian(\frac{1}{4})$.*

*Proof.* For all $\lambda \in \mathbb{R}$, we have

$$\ln(\mathbb{E}(e^{\lambda(x-p)}) = \ln\left(pe^{\lambda(1-p)} + (1-p)e^{-p\lambda}\right) = -p\lambda + \ln(1 + pe^{\lambda} - p) \leq \frac{\lambda^2}{8}.$$

Therefore $\mathbb{E}(e^{\lambda(x-p)}) \leq e^{\lambda^2(1/4)/2}$. □

### A.2 Proof of Proposition 1

*Proof.* Let $\mathbb{P}_\Theta$ denotes the (either Gaussian or Bernoulli) tensor block model, where $\Theta = \mathcal{C} \times_1 M_1 \times_2 \cdots \times_K M_K$ parameterizes the mean tensor. Since the mapping $\Theta \mapsto \mathbb{P}_\Theta$ is one-to-one, $\Theta$ is identifiable. Now suppose that $\Theta$ can be decomposed in two ways, $\Theta = \Theta(\{M_k\}, \mathcal{C}) = \Theta(\{\tilde{M}_k\}, \tilde{\mathcal{C}})$. Based on the Assumption 1, we have

$$\Theta = \mathcal{C} \times_1 M_1 \times_2 \cdots \times_K M_K = \tilde{\mathcal{C}} \times_1 \tilde{M}_1 \times_2 \cdots \times_K \tilde{M}_K, \tag{1}$$

where $\mathcal{C}, \tilde{\mathcal{C}} \in \mathbb{R}^{R_1 \times \cdots \times R_K}$ are two irreducible cores, and $M_k, \tilde{M}_k \in \{0,1\}^{R_k \times d_k}$ are membership matrices for all $k \in [K]$. We will prove by contradiction that $M_k$ and $\tilde{M}_k$ induce the same partition of $[d_k]$, for all $k \in [K]$.

Suppose the above claim does not hold. Then there exists a mode $k \in [K]$ such that the $M_k, \tilde{M}_k$ induce two different partitions of $[d_k]$. Without loss of generality, we assume $k = 1$. The definition of partition implies that there exists a pair of indices $i \neq j$, $i, j \in [d_1]$, such that, $i, j$ belong to the same cluster based on $M_1$, but they belong to different clusters based on $\tilde{M}_1$. Let $\mathcal{A} \neq \mathcal{B}, \mathcal{A}, \mathcal{B} \subset [d_1]$ respectively denote the clusters that $i$ and $j$ belong to, based on $\tilde{M}_1$. The left-hand side of (1) implies

$$\Theta_{i,i_2,\ldots,i_K} = \Theta_{j,i_2,\ldots,i_K}, \quad \text{for all } (i_2, \ldots, i_K) \in [d_2] \times \cdots \times [d_K]. \tag{2}$$

On the other hand, (1) implies

$$\Theta_{i,i_2,\ldots,i_K} = \Theta_{k,i_2,\ldots,i_K}, \quad \text{for all } k \in \mathcal{A} \text{ and all } (i_2, \ldots, i_K) \in [d_2] \times \cdots \times [d_K], \tag{3}$$

and

$$\Theta_{j,i_2,\ldots,i_K} = \Theta_{k,i_2,\ldots,i_K}, \quad \text{for all } k \in \mathcal{B} \text{ and all } (i_2, \ldots, i_K) \in [d_2] \times \cdots \times [d_K]. \tag{4}$$

Combining (2), (3) and (4), we have

$$\Theta_{i,i_2,\ldots,i_K} = \Theta_{k,i_2,\ldots,i_K}, \quad \text{for all } k \in \mathcal{A} \cup \mathcal{B} \text{ and all } (i_2, \ldots, i_K) \in [d_2] \times \cdots \times [d_K]. \tag{5}$$

Equation (5) implies that $\mathcal{A}$ and $\mathcal{B}$ can be merged into one cluster. This contradicts the irreducibility assumption of the core tensor $\tilde{\mathcal{C}}$. Therefore, $M_1$ and $\tilde{M}_1$ induce a same partition of $[d_1]$, and thus they are equal up to permutation of cluster labels. The proof is now complete. □

## A.3 Proof of Theorem 1

The following lemma is useful for the proof of Theorem 1.

**Lemma 1.** *Suppose $\mathcal{Y} = \Theta_{true} + \mathcal{E}$ with $\Theta_{true} \in \mathcal{P}$. Let $\hat{\Theta} = \arg\min_{\Theta \in \mathcal{P}} \|\hat{\Theta} - \mathcal{Y}\|_F^2$ be the least-square estimator of $\Theta_{true}$. We have*

$$\|\hat{\Theta} - \Theta_{true}\|_F \leq 2 \sup_{\mu \in \frac{\mathcal{P} - \mathcal{P}'}{|\mathcal{P} - \mathcal{P}'|}} \langle \mu, \mathcal{E} \rangle,$$

*where $\mathcal{P} - \mathcal{P}' = \{\Theta - \Theta' \colon \Theta, \Theta' \in \mathcal{P}\}$ and $\mathcal{S}/|\mathcal{S}| = \{s/\|s\|_2 \colon s \in \mathcal{S}\}$.*

*Proof.* Based on the definition of least-square estimator, we have

$$\|\hat{\Theta} - \mathcal{Y}\|_F^2 \leq \|\Theta_{\text{true}} - \mathcal{Y}\|_F^2. \tag{6}$$

Combining (6) with the fact

$$
\begin{aligned}
\|\hat{\Theta} - \mathcal{Y}\|_F^2 &= \|\hat{\Theta} - \Theta_{\text{true}} + \Theta_{\text{true}} - \mathcal{Y}\|_F^2 \\
&= \|\hat{\Theta} - \Theta_{\text{true}}\|_F^2 + \|\Theta_{\text{true}} - \mathcal{Y}\|_F^2 + 2\langle \hat{\Theta} - \Theta_{\text{true}}, \Theta_{\text{true}} - \mathcal{Y} \rangle,
\end{aligned}
$$

yields

$$\|\hat{\Theta} - \Theta_{\text{true}}\|_F^2 \leq 2\langle \hat{\Theta} - \Theta_{\text{true}}, \mathcal{Y} - \Theta_{\text{true}} \rangle = 2\langle \hat{\Theta} - \Theta_{\text{true}}, \mathcal{E} \rangle.$$

Dividing each side by $\|\hat{\Theta} - \Theta_{\text{true}}\|_F$, we have

$$\|\hat{\Theta} - \Theta_{\text{true}}\|_F \leq 2 \left\langle \frac{\hat{\Theta} - \Theta_{\text{true}}}{\|\hat{\Theta} - \Theta_{\text{true}}\|_F}, \mathcal{E} \right\rangle.$$

The desired inequality follows by noting $\frac{\hat{\Theta} - \Theta_{\text{true}}}{\|\hat{\Theta} - \Theta_{\text{true}}\|_F} \in \frac{\mathcal{P} - \mathcal{P}'}{|\mathcal{P} - \mathcal{P}'|}$. □

*Proof of Theorem 1.* To study the performance of the least-square estimator $\hat{\Theta}$, we need to introduce some additional notation. We view the membership matrix $M_k$ as an onto function $M_k \colon [d_k] \mapsto [R_k]$. With a little abuse of notation, we still use $M_k$ to denote the mapping function and write $M_k \in R_k^{d_k}$ by convention. We use $M = \{M_k\}_{k \in [K]}$ to denote the collection of $K$ membership matrices, and write $\mathcal{M} = \{M \colon M \text{ is the collection of membership matrices } M_k\text{'s}\}$. For any set $J$, $|J|$ denotes its cardinality. Note that $|\mathcal{M}| \leq \prod_k R_k^{d_k}$, because each $M_k$ can be identified by a partition of $[d_k]$ into $R_k$ disjoint non-empty sets.

For ease of notation, we define $d = \prod_k d_k$ and $R = \prod_k R_k$. We sometimes identify a tensor in $\mathbb{R}^{d_1 \times \cdots \times d_K}$ with a vector in $\mathbb{R}^d$. By the definition of the parameter space $\mathcal{P}$, the element $\Theta \in \mathcal{P}$ can be equivalently identified by $\Theta = \Theta(M, C)$, where $M \in \mathcal{M}$ is the collection of $K$ membership matrices and $C = \text{vec}(\mathcal{C}) \in \mathbb{R}^R$ is the core tensor. Note that, for a fixed clustering structure $M$, the space consisting of $\Theta = \Theta(M, \cdot)$ is a linear space of dimension $R$.

Now consider the least-square estimator

$$\hat{\Theta} = \arg\min_{\Theta \in \mathcal{P}} \{-2\langle \mathcal{Y}, \Theta \rangle + \|\Theta\|_F^2\} = \arg\min_{\Theta \in \mathcal{P}} \{\|\mathcal{Y} - \Theta\|_F^2\}.$$

Based on the Lemma 1,

$$
\begin{aligned}
\|\hat{\Theta} - \Theta_{\text{true}}\|_F &\leq 2 \sup_{\Theta \in \mathcal{P}} \sup_{\Theta' \in \mathcal{P}} \left\langle \frac{\Theta - \Theta'}{\|\Theta - \Theta'\|_F}, \mathcal{E} \right\rangle \\
&\leq 2 \sup_{M, M' \in \mathcal{M}} \sup_{C, C' \in \mathbb{R}^R} \left\langle \frac{\Theta(M, C) - \Theta'(M', C')}{\|\Theta(M, C) - \Theta'(M', C')\|_F}, \mathcal{E} \right\rangle.
\end{aligned}
$$

By union bound, we have, for any $t > 0$,

$$
\begin{aligned}
\mathbb{P}\left(\|\hat{\Theta} - \Theta_{\text{true}}\|_F > t\right) &\leq \mathbb{P}\left(\sup_{M,M'\in\mathcal{M}}\sup_{C,C'\in\mathbb{R}^R}\left|\left\langle\frac{\Theta(M,C)-\Theta'(M',C')}{\|\Theta(M,C)-\Theta'(M',C')\|_F}, \mathcal{E}\right\rangle\right| > \frac{t}{2}\right) \\
&\leq \sum_{M,M'\in\mathcal{M}}\mathbb{P}\left(\sup_{C'\in\mathbb{R}^R}\sup_{C\in\mathbb{R}^R}\left|\left\langle\frac{\Theta(M,\mathcal{C})-\Theta'(M',\mathcal{C})}{\|\Theta(M,\mathcal{C})-\Theta'(M',\mathcal{C})\|_F}, \mathcal{E}\right\rangle\right| \geq \frac{t}{2}\right) \\
&\leq |\mathcal{M}|^2 C_1^R \exp\left(-\frac{C_2 t^2}{32\sigma^2}\right) \\
&= \exp\left(2\sum_k d_k \log R_k + C_1 \prod_k R_k - \frac{C_2 t^2}{32\sigma^2}\right),
\end{aligned}
$$

for two universal constants $C_1, C_2 > 0$. Here the third line follows from [1] (Theorem 1.19) and the fact that $\Theta = \Theta(M, \cdot)$ lies in a linear space of dimension $R$. The last line uses $|\mathcal{M}| \leq \prod_k R_k^{d_k}$ and $R = \prod_k R_k$. Choosing $t = C\sigma\sqrt{\prod_k R_k + \sum_k d_k \log R_k}$ yields the desired bound. □

## A.4 Proof of Theorem 2

First we give a list of notation used in the proof. For ease of notation, we allow the basic arithmetic operators $(+, -, \geq, \text{etc})$ to be applied to pairs of vectors in an element-wise manner.

### A.4.1 Notations

$M_k = [\![m_{ir}^{(k)}]\!] \in \{0,1\}^{d_k \times R_k}$: the mode-$k$ membership matrix. The element $m_{ir}^{(k)} = 1$ if and only if the $i$th slide in mode $k$ belongs to the $r$th cluster.

$M_{k,\text{true}}, \hat{M}_k \in \{0,1\}^{d_k \times R_k}$: the true and estimated mode-$k$ cluster membership matrices, respectively.

$p^{(k)} = [\![p_r^{(k)}]\!] \in [0,1]^{R_k}$: the marginal cluster proportion vector listing the relative cluster sizes along the mode $k$. The element $p_r^{(k)} = \frac{1}{d_k}\sum_{i=1}^{d_k} \mathbb{1}\{m_{ir}^{(k)} = 1\}$ denotes the proportion of the $r$th cluster. The cluster proportion vector $p^{(k)} = p^{(k)}(M_k)$ can be viewed as a function of $M_k$.

$p_{\text{true}}^{(k)}, \hat{p}^{(k)} \in [0,1]^{R_k}$: the true and estimated mode-$k$ cluster proportion vectors, respectively.

$D^{(k)} = [\![D_{rr'}^{(k)}]\!] \in [0,1]^{R_k \times R_k}$: the mode-$k$ confusion matrix between clustering $M_{k,\text{true}}$ and $\hat{M}_k$. The entries in the confusion matrix is $D_{rr'}^{(k)} = \frac{1}{d_k}\sum_{i=1}^{d_k} \mathbb{I}\{m_{ir,\text{true}}^{(k)} = \hat{m}_{ir'}^{(k)} = 1\}$. The confusion matrix $D^{(k)} = \frac{1}{d_k}M_{k,\text{true}}^T \hat{M}_k$ is a function of $M_{k,\text{true}}$ and $\hat{M}_k$.

$\mathcal{J}_\tau = \{(M_1, \ldots, M_K) : p^{(k)}(M_k) \geq \tau \text{ for all } k \in [K]\}$: the set of all possible partitions that satisfy the marginal non-degenerating assumption.

$\mathcal{I} \subset 2^{[d_1]} \times \cdots \times 2^{[d_K]}$: the set of blocks that satisfy the marginal non-degenerating assumption for all $k \in [K]$;

$L = \inf\{|I| : I \in \mathcal{I}\}$: the minimum block size in $\mathcal{I}$.

$\|\mathcal{A}\|_{\max} = \max_{r_1,\ldots,r_K} |a_{r_1,\ldots,r_K}|$ for any tensor $\mathcal{A} = [\![a_{i_1,\ldots,i_K}]\!] \in \mathbb{R}^{R_1 \times \ldots \times R_K}$.

$f(x) = x^2$: the quadratic objective function.

**Remark 1.** *By definition, the confusion matrix $D^{(k)}$ satisfies the following two properties:*

1. *$D^{(k)}\mathbf{1} = p_{\text{true}}^{(k)}, (D^{(k)})^T\mathbf{1} = \hat{p}^{(k)}$.*

2. *The estimated clustering matches the true clustering if and only if $D^{(k)}$ equals to the diagonal matrix up to permutation.*

### A.4.2 Auxiliary Results

Recall that the objective function in our tensor block model is

$$f(\mathcal{C}, \{\boldsymbol{M}_k\}) = \langle \mathcal{Y}, \ \Theta \rangle - \frac{\|\Theta\|_F^2}{2}, \tag{7}$$
$$\text{where } \Theta = \mathcal{C} \times_1 \boldsymbol{M}_1 \times_2 \cdots \times_K \boldsymbol{M}_K,$$

where $\mathcal{Y} \in \mathbb{R}^{d_1 \times \cdots \times d_K}$ is the data, $\mathcal{C}$ is the core tensor of interest, and $\{\boldsymbol{M}_k\}$ is the membership matrices of interest. Without loss of generality, we will work with the scaled objective $\frac{2}{\prod_k d_k} f(\mathcal{C}, \{\boldsymbol{M}_k\})$. With a little abuse of notation, we still denote the scaled function as $f(C, \{\boldsymbol{M}_k\})$.

We will prove that, if there is non-negligible mismatch between $\{\hat{\boldsymbol{M}}_k\}$ and $\{\boldsymbol{M}_{k,\text{true}}\}$, then $\{\hat{\boldsymbol{M}}_k\}$ cannot be the optimizer to (7). To show this, we investigate the objective values at the global optimizer vs. at the true parameter. The deviation between these two values comes from two aspects: the label assignments (i.e., the estimation of $\{\boldsymbol{M}_k\}$) and the estimation of the core tensor. In what follows, we tease apart these two aspects.

1. First, suppose the partitions $\{\boldsymbol{M}_k\}$ are given, which are not necessarily equal to $\{\boldsymbol{M}_{k,\text{true}}\}$. We now assess the stochastic error due to estimation of $\mathcal{C}$, conditional on $\{\boldsymbol{M}_k\}$. In such a case, the core $\hat{\mathcal{C}} = \arg\min_{\mathcal{C}} f(\mathcal{C}, \{\boldsymbol{M}_k\})$ can be solved explicitly. Specifically, the optimizer $\hat{\mathcal{C}} = [\![\hat{c}_{r_1,\dots,r_K}]\!]$ consists of the sample averages of each tensor block, where

$$\begin{aligned} \hat{c}_{r_1,\dots,r_K} &= \hat{c}_{r_1,\dots,r_K}(\{\boldsymbol{M}_k\}) \\ &= \frac{1}{d_1 \cdots d_K} \frac{1}{p_{r_1}^{(1)} \cdots p_{r_K}^{(K)}} \left[ \mathcal{Y} \times_1 \boldsymbol{M}_1^T \times_2 \cdots \times_K \boldsymbol{M}_K^T \right]_{r_1,\dots,r_K} \end{aligned} \tag{8}$$

where the marginal cluster proportion $p_{r_k}^{(k)}$ is induced by the clustering $\boldsymbol{M}_k$.

Define a new cost function $F(\boldsymbol{M}_1, \dots, \boldsymbol{M}_K) = -f(\hat{\mathcal{C}}, \boldsymbol{M}_1, \dots, \boldsymbol{M}_K)$, where $\hat{\mathcal{C}} = [\![\hat{c}_{i_1,\dots,i_K}]\!]$ is expressed in (8). A straightforward calculation shows that the function $F(\cdot)$ has the form

$$F(\boldsymbol{M}_1, \dots, \boldsymbol{M}_K) = \sum_{r_1,\dots,r_K} \left( \prod_k p_{r_k}^{(k)} \right) \hat{c}_{r_1,\dots,r_K}^2. \tag{9}$$

Let $G(\boldsymbol{M}_1, \dots, \boldsymbol{M}_k) = \mathbb{E}(F(\boldsymbol{M}_1, \dots, \boldsymbol{M}_K))$, where the expectation is taken with respect to the $\hat{\mathcal{C}} = [\![\hat{c}_{r_1,\dots,r_K}]\!]$. We have that

$$G(\boldsymbol{M}_1, \dots, \boldsymbol{M}_K) = \sum_{r_1,\dots,r_K} \left( \prod_k p_{r_k}^{(k)} \right) \mu_{r_1,\dots,r_K}^2, \tag{10}$$

where

$$\mu_{r_1,\dots,r_K} = \mathbb{E}(\hat{c}_{r_1,\dots,r_K}) = \frac{1}{\prod_k p_{r_k}^{(k)}} \left[ \mathcal{C} \times_1 \boldsymbol{D}^{(1)^T} \times_2 \cdots \times_K \boldsymbol{D}^{(K)^T} \right]_{r_1,\dots,r_K}$$

is the expectation of the average of $y_{i_1,\dots,i_K}$ over the tensor block indexed by $(r_1, \dots, r_K)$, and $\boldsymbol{D}^{(k)} = [\![D_{i_k j_k}^{(k)}]\!]$ is the confusion matrix between $\boldsymbol{M}_{k,\text{true}}$ and $\boldsymbol{M}_k$.

The deviation $F(\boldsymbol{M}_1, \dots, \boldsymbol{M}_K) - G(\boldsymbol{M}_1, \dots, \boldsymbol{M}_K)$ quantifies the stochastic error caused by the core tensor estimation. We sometimes use $G(\boldsymbol{D}^{(1)}, \dots, \boldsymbol{D}^{(K)})$ to denote $G(\boldsymbol{M}_1, \dots, \boldsymbol{M}_k)$ if we want to emphasize the error caused by mismatch in label assignments. Based on (9) and (10), we define a residual tensor for the block means:

$$\begin{aligned} \mathcal{R}(\boldsymbol{M}_1, \dots, \boldsymbol{M}_K) &= [\![R_{r_1,\dots,r_K}]\!], \text{ where} \\ R_{r_1,\dots,r_K} &= \hat{c}_{r_1,\dots,r_K} - \mu_{r_1,\dots,r_K}, \quad \text{for all } (r_1, \dots, r_K) \in [R_1] \times \cdots \times [R_K]. \end{aligned} \tag{11}$$

Note that, conditional on $\{\boldsymbol{M}_k\}$, the entries $R_{r_1,\dots,r_K}$ in the residual tensor are independent sub-Gaussian with parameter depending on the size of the $(r_1, \dots, r_K)$th block.

2. Next, we free $\{M_k\}$ and quantify the total stochastic deviation. Note that optimizing (7) is equivalent to optimizing (9) with respect to $\{M_k\}$. So the least-square estimator of $\{M_k\}$ can be expressed as

$$(\hat{M}_1, \ldots, \hat{M}_K) = \underset{(M_1,\ldots,M_K)\in\mathcal{J}_\tau}{\arg\max} F(M_1, \ldots, M_K).$$

The expectation (with respect to $\hat{\mathcal{C}}$) of the objective value at the true parameter is

$$G(M_{1,\text{true}}, \ldots, M_{K,\text{true}}) = \sum_{r_1,\ldots,r_K} p^{(1)}_{r_1,\text{true}} \cdots p^{(K)}_{r_K,\text{true}} c^2_{r_1,\ldots,r_K,\text{true}}$$

We use $G(D^{(1)}, \ldots, D^{(K)}) - G(M_{1,\text{true}}, \ldots, M_{K,\text{true}})$ to measure the stochastic deviation caused by mismatch in label assignments; and use $F(M_1, \ldots, M_K) - G(D^{(1)}, \ldots, D^{(K)})$ to measure stochastic deviation caused by estimation of core tensors.

The following lemma shows that, if there is non-negligible mismatch between $M_{k,\text{true}}$ and $\hat{M}_k$, then $\hat{M}_k$ cannot be the global optimizer to the objective function (7).

**Lemma 2.** *Consider partitions that satisfying $(M_1, \ldots, M_K) \in \mathcal{J}_\tau$, for some $\tau > 0$. Define the minimal gap between block means $\delta^{(k)} = \min_{r_k \neq r'_k} \max_{r_1,\ldots,r_{k-1},r_{k+1},\ldots,r_K} (c_{r_1,\ldots,r_k,\ldots,r_K} - c_{r_1,\ldots,r'_k,\ldots,r_K})^2 > 0$ and assume $\delta_{\min} = \min_k \delta^{(k)} > 0$. For any fixed $\varepsilon > 0$, suppose $MCR(M_{k,\text{true}}, \hat{M}_k) \geq \varepsilon$ for some $k \in [K]$. Then, we have*

$$G(D^{(1)}, \ldots, D^{(K)}) - G(M_{1,\text{true}}, \ldots, M_{K,\text{true}}) \leq -\frac{1}{4}\varepsilon\tau^{K-1}\delta_{\min},$$

*where $D^{(k)}$ is the confusion matrix between $M_{k,\text{true}}$ and $\hat{M}_k$.*

*Proof of Lemma 2.* For ease of notation, we drop the subscript "true" and simply write $p^{(k)}_{r_k}$, $M_k$, $\mathcal{C}$, etc. as the true parameters. The corresponding estimators are denoted as $\hat{p}^{(k)}_{r_k}$, $\hat{M}_k$, etc. Recall that

$$G(D^{(1)}, \ldots, D^{(K)}) = \sum_{r_1,\ldots,r_K} \hat{p}^{(1)}_{r_1} \cdots \hat{p}^{(K)}_{r_K} \mu^2_{r_1,\ldots,r_K},$$

where $\hat{p}^{(k)}_{r_k}$ is the marginal cluster proportion induced by $\hat{M}_k$, and $\mu_{r_1,\ldots,r_K}$ is the expected block mean induced by $\hat{M}_k$:

$$\mu_{r_1,\ldots,r_K} = \mu_{r_1,\ldots,r_K}(\hat{M}_1, \ldots, \hat{M}_K) = \frac{1}{\prod_k \hat{p}^{(k)}_{r_k}} \left[ \mathcal{C} \times_1 D^{(1)^T} \times_2 \cdots \times_K D^{(K)^T} \right]_{r_1,\ldots,r_K}.$$

We provide the proof for $k = 1$. The proof for other $k \in [K]$ is similar. The condition on MCR implies that, there exist some $r_1 \in [R_1]$ and some $a_1 \neq a'_1 \in [R_1]$, such that $\min\{D^{(1)}_{a_1 r_1}, D^{(1)}_{a'_1 r_1}\} \geq \varepsilon$. Because the minimal gap between tensor block means are non-zero, we choose $(a_2, \ldots, a_K)$ such that $(c_{a_1,a_2,\ldots,a_K} - c_{a'_1,a_2,\ldots,a_K})^2 = \max_{a_2,\ldots,a_K} (c_{a_1,a_2,\ldots,a_K} - c_{a'_1,a_2,\ldots,a_K})^2 > 0$.

Let $\mathcal{N} = [\![c^2_{a_1,\ldots,a_K}]\!] \in \mathbb{R}^{R_1 \times \cdots \times R_K}$ be the quadratic loss evaluated at block, $W_{r_1,\ldots,r_K} = \prod_k \hat{p}^{(k)}_{r_k} > 0$ the size for the block indexed by $(r_1, \ldots, r_K)$. For ease of notation, we drop the subscript $(r_1, \ldots, r_K)$ and simply write $W$.

Based on the convexity of quadratic loss, there exists $c_* \in \mathbb{R}$ such that the weighted quadratic loss can be expressed as

$$[\mathcal{N} \times_1 D^{(1)^T} \times_2 \cdots \times_K D^{(K)^T}]_{r_1,\ldots,r_K}$$
$$= D^{(1)}_{a_1 r_1} D^{(2)}_{a_2 r_2} \cdots D^{(K)}_{a_K r_K} c^2_{a_1,a_2,\ldots,a_K} + D^{(1)}_{a'_1 r_1} D^{(2)}_{a_2 r_2} \cdots D^{(K)}_{a_K r_K} c^2_{a'_1,a_2,\ldots,a_K} +$$
$$(W - D^{(1)}_{a_1 r_1} D^{(1)}_{a_2 r_2} \cdots D^{(K)}_{a_K r_K} - D^{(1)}_{a'_1 r_1} D^{(1)}_{a_2 r_2} \cdots D^{(K)}_{a_K r_K})c^2_*.$$

Recall that $\mu_{r_1,\ldots,r_K} = \frac{1}{W}[\mathcal{C} \times_1 \boldsymbol{D}^{(1)^T} \times_2 \cdots \times_K \boldsymbol{D}^{(K)^T}]_{r_1,\ldots,r_K}$ is the $(r_1,\ldots,r_k)$-th weighted entry of the block means. By the Taylor expansion of quadratic loss function at $\mu_{r_1,\ldots,r_K}$, we have

$$\frac{1}{W}[\mathcal{N} \times_1 \boldsymbol{D}^{(1)^T} \times_2 \cdots \times_K \boldsymbol{D}^{(K)^T}]_{r_1,\ldots,r_K} - \mu^2_{r_1,\ldots,r_K}$$

$$\geq \frac{1}{2W} D^{(1)}_{a_1 r_1} D^{(2)}_{a_2 r_2} \cdots D^{(K)}_{a_K r_K} (c_{a_1,a_2,\ldots,a_K} - \mu_{r_1,\ldots,r_K})^2 +$$

$$\frac{1}{2W} D^{(1)}_{a'_1,r_1} D^{(2)}_{a_2 r_2} \cdots D^{(K)}_{a_K r_K} (c_{a'_1,a_2,\ldots,a_K} - \mu_{r_1,\ldots,r_K})^2 +$$

$$\frac{1}{2W}\left(W - D^{(1)}_{a_1 r_1} D^{(2)}_{a_2,r_2} \cdots D^{(K)}_{a_K r_K} - D^{(1)}_{a'_1,r_1} D^{(2)}_{a_2,r_2} \cdots D^{(K)}_{a_K r_K}\right)(c_* - \mu_{r_1,\ldots,r_K})^2. \quad (12)$$

Combining (12) and basic inequality $(a^2 + b^2) \geq \frac{1}{2}(a+b)^2$ gives

$$\frac{1}{W}[\mathcal{N} \times_1 \boldsymbol{D}^{(1)^T} \times_2 \cdots \times_K \boldsymbol{D}^{(K)^T}]_{r_1,\ldots,r_K} - \mu^2_{r_1,\ldots,r_K}$$

$$\geq \frac{1}{4W} \min\left\{D^{(1)}_{a_1 r_1}, D^{(1)}_{a'_1 r_1}\right\} D^{(2)}_{a_2 r_2} \cdots D^{(K)}_{a_K r_K} (c_{a_1,\ldots,a_K} - c_{a'_1,\ldots,a_K})^2$$

$$\geq \frac{\varepsilon D^{(2)}_{a_2 r_2} \cdots D^{(K)}_{a_K r_K}}{4W}(c_{a_1,a_2,\ldots,a_K} - c_{a'_1,a_2,\ldots,a_K})^2. \quad (13)$$

The inequality (13) only holds for a certain $r_1 \in [R_1]$. For any other $r'_1 \in [R_1]/\{r_1\}$, by Jensen's inequality we have

$$\frac{1}{W}[\mathcal{N} \times_1 \boldsymbol{D}^{(1)^T} \times_2 \cdots \times_K \boldsymbol{D}^{(K)^T}]_{r'_1,\ldots,r_K} - \mu^2_{r'_1,\ldots,r_K} \geq 0. \quad (14)$$

Combining the sum of (13) and (14) over $(r_2,\ldots,r_K)$ gives

$$G(\boldsymbol{D}^{(1)},\ldots,\boldsymbol{D}^{(K)}) - \sum_{r_1,\ldots,r_K} p^{(1)}_{r_1} \cdots p^{(K)}_{r_K} c^2_{r_1,\ldots,r_K}$$

$$\leq -\varepsilon \sum_{r_2,\ldots,r_K} \frac{D^{(2)}_{a_2 r_2} \cdots D^{(K)}_{a_K r_K}}{4}(c_{a_1,a_2,\ldots,a_K} - c_{a'_1,a_2,\ldots,a_K})^2$$

$$\leq -\frac{1}{4}\varepsilon \tau^{K-1} \delta_{\min},$$

where the last line uses the fact that $\sum_{r_k} D^{(k)}_{a_k r_k} = p^{(k)}_{a_k} \geq \tau$. $\qquad\square$

### A.4.3 Proof

*Proof of Theorem 2.* The notations we use here are inherited from Lemma 2. With a little abuse of notation, we use $\delta_{\min} = \min_k \delta^{(k)}$ in the proof. This differs from the definition $\delta_{\min} = \frac{1}{\|\mathcal{C}\|_{\max}} \min_k \delta^{(k)}$ in Theorem 2 by a factor of $\|\mathcal{C}\|_{\max}$. By Lemma 2, we obtain that

$$\mathbb{P}\left(\mathrm{MCR}(\hat{\boldsymbol{M}}_k, \boldsymbol{M}_{k,\mathrm{true}}) \geq \varepsilon\right)$$

$$\leq \mathbb{P}\left(G(\boldsymbol{D}^{(1)},\ldots,\boldsymbol{D}^{(K)}) - G(\boldsymbol{M}_{1,\mathrm{true}},\ldots,\boldsymbol{M}_{K,\mathrm{true}}) \leq -\frac{1}{4}\varepsilon \tau^{K-1} \delta_{\min}\right). \quad (15)$$

Define $r = \sup_{\mathcal{J}_\tau} |F(\boldsymbol{M}_1,\ldots,\boldsymbol{M}_K) - G(\boldsymbol{D}^{(1)},\ldots,\boldsymbol{D}^{(K)})|$ as the stochastic deviation caused by the label assignment. When the event $G(\boldsymbol{D}^{(1)},\ldots,\boldsymbol{D}^{(K)}) - G(\boldsymbol{M}_{1,\mathrm{true}},\ldots,\boldsymbol{M}_{K,\mathrm{true}}) \leq -\frac{1}{4}\varepsilon\tau^{K-1}\delta_{\min}$ holds, by triangle inequality, we have

$$F(\hat{\boldsymbol{M}}_1,\ldots,\hat{\boldsymbol{M}}_K) - F(\boldsymbol{M}_{1,\mathrm{true}},\ldots,\boldsymbol{M}_{K,\mathrm{true}}) \leq 2r - \frac{1}{4}\varepsilon\tau^{K-1}\delta_{\min}. \quad (16)$$

Plugging the event (16) back into inequality (15), we obtain

$$\mathbb{P}\left(\mathrm{MCR}(\hat{\boldsymbol{M}}_k, \boldsymbol{M}_{k,\mathrm{true}}) \geq \varepsilon\right)$$

$$\leq \mathbb{P}\left(F(\hat{\boldsymbol{M}}_1, \ldots, \hat{\boldsymbol{M}}_K) - F(\boldsymbol{M}_{1,\mathrm{true}}, \ldots, \boldsymbol{M}_{K,\mathrm{true}}) \leq 2r - \frac{1}{4}\varepsilon\tau^{K-1}\delta_{\min}\right)$$

$$\leq \mathbb{P}\left(r \geq \frac{\varepsilon\tau^{K-1}\delta_{\min}}{8}\right), \tag{17}$$

where the last line uses the fact that the $\hat{\boldsymbol{M}}_k$ is the global optimizer of $F(\cdot)$; i.e. $F(\hat{\boldsymbol{M}}_1, \ldots, \hat{\boldsymbol{M}}_K) = \arg\max F(\boldsymbol{M}_1, \ldots, \boldsymbol{M}_K) \geq F(\boldsymbol{M}_{1,\mathrm{true}}, \ldots, \boldsymbol{M}_{K,\mathrm{true}})$.

Now we aim to find the probability (17) with respect to $r = \sup_{\mathcal{J}_\tau}|F(\boldsymbol{M}_1, \ldots, \boldsymbol{M}_K) - G(\boldsymbol{D}^{(1)}, \ldots, \boldsymbol{D}^{(K)})|$. Note that $r$ involves the quadratic objective $f(x) = x^2$. The quadratic function $f(x)$ is locally lipschitz continuous with lipschitz constant $b = \sup_x|f'(x)|$, where $x$ is in the closure of the convex hull of the entries of $\mathcal{C}$. Note that $b \leq 2\|\mathcal{C}\|_{\max}$. Therefore, for any partitions $\{\boldsymbol{M}_k\}$ (which are not necessarily equal to $\{\hat{\boldsymbol{M}}_k\}$ or $\{\boldsymbol{M}_{k,\mathrm{true}}\}$):

$$\left|F(\boldsymbol{M}_1, \ldots, \boldsymbol{M}_K) - G(\boldsymbol{D}^{(1)}, \ldots, \boldsymbol{D}^{(K)})\right|$$

$$\leq \sum_{r_1, \ldots, r_K} p_{r_1}^{(1)} p_{r_2}^{(2)} \cdots p_{r_K}^{(K)} |f(\hat{c}_{r_1, \ldots, r_K}) - f(\mu_{r_1, \ldots, r_K})|$$

$$\leq 2\|\mathcal{C}\|_{\max}\|\mathcal{R}(\boldsymbol{M}_1, \ldots, \boldsymbol{M}_K)\|_{\max}, \tag{18}$$

where

$$\hat{c}_{r_1, \ldots, r_K} = \frac{1}{\prod_k p_{r_k}^{(k)}} (\mathcal{Y} \times_1 \boldsymbol{M}_1^T \times_2 \cdots \times_K \boldsymbol{M}_K^T)_{r_1, \ldots, r_K},$$

and

$$\mu_{r_1, \ldots, r_K} = \frac{1}{\prod_k p_{r_k}^{(k)}} \left[\mathcal{C} \times_1 \boldsymbol{D}^{(1)^T} \times_2 \cdots \times_K \boldsymbol{D}^{(K)^T}\right]_{r_1, \ldots, r_K}.$$

are, respectively, sample average and expected sample average, conditional on the partitions $\boldsymbol{M}_k$, and $\mathcal{R}(\boldsymbol{M}_1, \ldots, \boldsymbol{M}_K)$ is the residual tensor defined in (11).

Combining (17), (18) and Hoeffding's inequality, we have

$$\mathbb{P}\left(\mathrm{MCR}(\hat{\boldsymbol{M}}_k, \boldsymbol{M}_{k,\mathrm{true}}) \geq \varepsilon\right) \leq \mathbb{P}\left(\sup_{\mathcal{J}_\tau}\|\mathcal{R}(\boldsymbol{M}_1, \ldots, \boldsymbol{M}_K)\|_{\max} \geq \frac{\varepsilon\tau^{K-1}\delta_{\min}}{16\|\mathcal{C}\|_{\max}}\right)$$

$$\leq \mathbb{P}\left(\sup_{I \in \mathcal{I}} \frac{\left|\sum_{(i_1, \ldots, i_K) \in I}(Y_{i_1, \ldots, i_K} - \mathbb{E}(Y_{i_1, \ldots, i_K}))\right|}{|I|} \geq \frac{\varepsilon\tau^{K-1}\delta_{\min}}{16\|\mathcal{C}\|_{\max}}\right)$$

$$\leq 2^{1+\sum_k d_k}\exp\left(-\frac{\varepsilon^2\tau^{2(K-1)}\delta_{\min}^2 L}{512\sigma^2\|\mathcal{C}\|_{\max}^2}\right), \tag{19}$$

where the last line uses the sub-Gaussianness of the entries in the residual tensor (conditional on $\{\boldsymbol{M}_k\}$), and $L = \inf\{|I| : I \subset \mathcal{I}\} \geq \tau^K\prod_{k=1}^K d_k$ is introduced in Section A.4.1. Defining $C = \frac{1}{512}$ in (19) yields the desired conclusion.

$\square$

## A.5 Sparse estimator

**Lemma 3.** *Consider the regularized least-square estimation,*

$$\hat{\Theta}^{sparse} = \arg\min_{\Theta \in \mathcal{P}}\left\{\|\mathcal{Y} - \Theta\|_F^2 + \lambda\|\mathcal{C}\|_\rho\right\}, \tag{20}$$

*where $\mathcal{C} = [\![c_{r_1, \ldots, r_K}]\!] \in \mathbb{R}^{R_1 \times \cdots \times R_K}$ is the block-mean tensor, $\|\mathcal{C}\|_\rho$ is the penalty function with $\rho$ being an index for the tensor norm, and $\lambda$ is the penalty tuning parameter. We have*

$$\hat{c}_{r_1, \ldots, r_K}^{sparse} = \begin{cases} \hat{c}_{r_1, \ldots, r_K}^{ols}\mathbb{1}\left\{|\hat{c}_{r_1, \ldots, r_K}^{ols}| \geq \sqrt{\frac{\lambda}{n_{r_1, \ldots, r_K}}}\right\} & if\ \rho = 0, \\ sign(\hat{c}_{r_1, \ldots, r_K}^{ols})\left(|\hat{c}_{r_1, \ldots, r_K}^{ols}| - \frac{\lambda}{2n_{r_1, \ldots, r_K}}\right)_+ & if\ \rho = 1, \end{cases} \tag{21}$$

where $a_+ = \max(a, 0)$ and $\hat{c}^{ols}_{r_1,\ldots,r_K}$ denotes the ordinary least-square estimate as in Algorithm 1.

*Proof.* We formulate the estimation of $\mathcal{C}$ as a regularized least-square regression. Note that $\Theta \in \mathcal{P}$ implies that

$$\Theta = \mathcal{C} \times_1 \boldsymbol{M}_1 \times \cdots \times_K \boldsymbol{M}_K.$$

Define $\boldsymbol{X} = \boldsymbol{M}_1 \otimes \ldots \otimes \boldsymbol{M}_K \in \mathbb{R}^{d \times R}$, where $d = \prod_k d_k$ and $R = \prod_k R_k$, and $\boldsymbol{\beta} = \text{vec}(\mathcal{C}) \in \mathbb{R}^R$. Here $\boldsymbol{X}$ is a membership matrix that indicates the block allocation among tensor entries. Specifically, $\boldsymbol{X}$ consists of orthogonal columns with $\boldsymbol{X}^T \boldsymbol{X} = \text{diag}(n_1, \ldots, n_R)$, where $n_r$ is the number of entries in the tensor block that corresponds to the $r$-th column of $\boldsymbol{X}$.

For a given set of $\boldsymbol{M}'_k s$, the optimization (22) with respect to $\mathcal{C}$ is equivalent to a regularized linear regression with $\boldsymbol{Y} = \text{vec}(\mathcal{Y})$ as the response and $\boldsymbol{X}$ as the design matrix:

$$L(\boldsymbol{\beta}) = \|\boldsymbol{Y} - \boldsymbol{X}\boldsymbol{\beta}\|_2^2 + \lambda\|\boldsymbol{\beta}\|_\rho. \tag{22}$$

When $\lambda = 0$ (no penalty), the minimizer is $\hat{\boldsymbol{\beta}}^{\text{ols}} = (\hat{\beta}_1^{\text{ols}}, \ldots, \hat{\beta}_R^{\text{ols}}) = (\boldsymbol{X}^T\boldsymbol{X})^{-1}\boldsymbol{X}^T\boldsymbol{Y}$, where $\hat{\beta}_r^{\text{ols}} = \frac{1}{n_r}\boldsymbol{y_r}\boldsymbol{1}_{n_r}^T$ for all $r \in [R]$.

**Case 1:** $\rho = 0$.

Note that $\boldsymbol{X}$ induces a partition of indices $[d]$ into $R$ blocks. With a little abuse of notation, we use $\mathcal{R} = \{i \in [d] : \boldsymbol{X}(i) = r\}$ to denote the collection of tensor indices that belong to the $r$th block, and use $\boldsymbol{Y}_\mathcal{R} \in \mathbb{R}^{n_r}$ to denote the corresponding tensor entries. By the orthogonality of $\boldsymbol{X}$, we have

$$L(\boldsymbol{\beta}) = \sum_{r=1}^R \|\boldsymbol{Y}_\mathcal{R} - \beta_r \boldsymbol{1}_{n_r}\|_2^2 + \lambda \sum_{r=1}^R \mathbb{1}\{\beta_r \neq 0\}$$

$$= \sum_{r=1}^R \underbrace{\left(\|\boldsymbol{Y}_\mathcal{R} - \beta_r \boldsymbol{1}_{n_r}\|_2^2 + \lambda\mathbb{1}\{\beta_r \neq 0\}\right)}_{:=L_r(\beta_r)}$$

The optimization can be separated into each of $\beta_r$'s. For any $r \in [R]$, the sub-optimization $\min_{\beta_r} L_r(\beta_r)$ has a closed-form solution

$$\min_{\beta_r} L_r(\beta_r) = \begin{cases} \boldsymbol{Y}_\mathcal{R}^T\boldsymbol{Y}_\mathcal{R} - n_r\left(\hat{\beta}_r^{\text{ols}}\right)^2 + \lambda & \text{if } \hat{\beta}_r^{\text{ols}} \neq 0, \\ \boldsymbol{Y}_\mathcal{R}^T\boldsymbol{Y}_\mathcal{R} & \text{if } \hat{\beta}_r^{\text{ols}} = 0, \end{cases}$$

with

$$\arg\min_{\beta_r} L_r(\beta_r) = \begin{cases} 0 & \text{if } n_r\left(\hat{\beta}_r^{\text{ols}}\right)^2 \leq \lambda, \\ \hat{\beta}_r^{\text{ols}} & \text{otherwise.} \end{cases} \tag{23}$$

Solution (23) can be simplified as $\hat{\beta}_r^{\text{sparse}} = \hat{\beta}_r^{\text{ols}}\mathbb{1}\{|\hat{\beta}_r^{\text{ols}}| \leq \sqrt{\frac{\lambda}{n_r}}\}$. The proof is complete by noting that $\hat{c}^{\text{sparse}}_{r_1,\ldots,r_R} = \hat{\beta}_r^{\text{sparse}}$ and $n_{r_1,\ldots,r_K} = n_r$ for all $(r_1, \ldots, r_K) \in [R_1] \times \cdots \times [R_K]$.

**Case 2:** $\rho = 1$.

Similar as in Case 1, we write the optimization (22) as

$$L(\boldsymbol{\beta}) = \sum_{r=1}^R \underbrace{\left(\|\boldsymbol{Y}_\mathcal{R} - \beta_r \boldsymbol{1}_{n_r}\|_2^2 + \lambda|\beta_r|\right)}_{:=L_r(\beta_r)},$$

where, with a little abuse of notation, we still use $L_r(\beta_r)$ to denote the sub-optimization. To solve $\arg\min_{\beta_r} L_r(\beta_r)$, we use the properties of subderivative. Taking the subderivative with respect to $\beta_r$, we obtain

$$\frac{\partial L_r(\beta_r)}{\partial \beta_r} = \begin{cases} 2n_r\beta_r - 2n_r\hat{\beta}_r^{\text{ols}} + \lambda & \text{if } \beta_r > 0, \\ [2n_r\beta_r - 2\hat{\beta}_r^{\text{ols}} - \lambda, \ 2n_r\beta_r - \hat{\beta}^{\text{ols}} + \lambda] & \text{if } \beta_r = 0, \\ 2n_r\beta_r - 2n_r\hat{\beta}_r^{\text{ols}} + \lambda & \text{if } \beta_r < 0. \end{cases}$$

Because $\hat{\beta}_r^{\text{sparse}}$ minimizes $L_r(\beta_r)$ if and only if $0 \in \frac{\partial L_r(\beta_r)}{\partial \beta_j}$, we have:

$$\hat{\beta}_r^{\text{sparse}} = \begin{cases} \hat{\beta}_r^{\text{ols}} + \frac{\lambda}{2n_r} & \text{if } \hat{\beta}_r^{\text{ols}} < -\frac{\lambda}{2n_r}, \\ 0 & \text{if } \hat{\beta}_r^{\text{ols}} \in [-\frac{\lambda}{2n_r}, \frac{\lambda}{2n_r}], \\ \hat{\beta}_r^{\text{ols}} - \frac{\lambda}{2n_r} & \text{if } \hat{\beta}_r^{\text{ols}} > \frac{\lambda}{2n_r}. \end{cases} \qquad (24)$$

The solution (24) can be simplified as

$$\hat{\beta}_r^{\text{sparse}} = \text{sign}(\hat{\beta}_r^{\text{ols}}) \left( |\hat{\beta}_r^{\text{ols}}| - \frac{\lambda}{2n_r} \right)_+, \quad \text{for all } r \in [R].$$

$\square$

## B Supplementary Figures and Tables

Figure S1: (a) estimation error and (b) sparse error rate against noise for sparse tensors of dimension $(40, 40, 40)$ when $p = 0.8$.

| Dimensions $(d_1, d_2, d_3)$ | True clustering sizes $(R_1, R_2, R_3)$ | Noise $(\sigma)$ | Estimated clustering sizes $(\hat{R}_1, \hat{R}_2, \hat{R}_3)$ |
|---|---|---|---|
| $(40, 40, 40)$ | $(4, 4, 4)$ | 4 | $(\mathbf{4},\ \mathbf{4},\ \mathbf{4}) \pm (0,\ 0,\ 0)$ |
| $(40, 40, 40)$ | $(4, 4, 4)$ | 8 | $(\mathbf{3.94},\ \mathbf{3.96},\ \mathbf{3.96}) \pm (0.03,\ 0.03,\ 0.03)$ |
| $(40, 40, 40)$ | $(4, 4, 4)$ | 12 | $(3.08,\ 3.12,\ 3.12) \pm (0.10, 0.10, 0.10)$ |
| $(40, 40, 80)$ | $(4, 4, 4)$ | 4 | $(\mathbf{4},\ \mathbf{4},\ \mathbf{4}) \pm (0,\ 0,\ 0)$ |
| $(40, 40, 80)$ | $(4, 4, 4)$ | 8 | $(\mathbf{4},\ \mathbf{4},\ \mathbf{4}) \pm (0,\ 0,\ 0)$ |
| $(40, 40, 80)$ | $(4, 4, 4)$ | 12 | $(\mathbf{3.96},\ \mathbf{3.96},\ 3.92) \pm (0.04, 0.04, 0.04)$ |
| $(40, 40, 40)$ | $(2, 3, 4)$ | 4 | $(\mathbf{2},\ \mathbf{3},\ \mathbf{4}) \pm (0,\ 0,\ 0)$ |
| $(40, 40, 40)$ | $(2, 3, 4)$ | 8 | $(\mathbf{2},\ \mathbf{3},\ \mathbf{3.96}) \pm (0,\ 0,\ 0.03)$ |
| $(40, 40, 40)$ | $(2, 3, 4)$ | 12 | $(\mathbf{2},\ \mathbf{2.96},\ 3.60) \pm (0,\ 0.05,\ 0.09)$ |

Table S1: The simulation results for estimating $\boldsymbol{R} = (R_1, R_2, R_3)$. Bold number indicates no significant difference between the estimate and the ground truth, based on a $z$-test with a level 0.05.

| Tissues | Over-expressed genes | Block-means | Under-expressed genes | Block-means |
|---|---|---|---|---|
| Cluster 1 | GFAP, MBP | 10.88 | GPR6 , DLX5 , DLX6 , NKX2-1 | -8.40 |
| Cluster 2 | GFAP, MBP | 5.98 | CDH9, RXFP1, CRH, ARX, CARTPT, DLX1,FEZF2 | -9.49 |
| Cluster 3 | GFAP, MBP | 8.34 | AVPR1A, CCKAR, CHRNB4, CYP19A1, HOXA4 , LBX1, SLC6A3 | -8.45 |
| | | | TBR1, SLC17A6, SLC30A3 | -8.17 |
| Cluster 4 | GFAP, MBP | 8.83 | AVPR1A, CCKAR, CHRNB4, CYP19A1, HOXA4 , LBX1, SLC6A3 | -8.40 |
| | | | DAO EN2 EOMES | -6.57 |

Table S2: Top expression blocks from the multi-tissue gene expression analysis. The tissue clusters are described in Supplementary Section D.

| Countries | Countries | Relation types |
|---|---|---|
| Cluster 1 | Clusters 4 and 5 | reltreaties, booktranslations, relbooktranslations, relexports, exports3 |
| Clusters 1 and 4 | Cluster 5 | relintergovorgs, relngo, intergovorgs3, ngoorgs3 |
| Cluster 3 | Clusters 1, 4, and 5 | commonbloc0, blockpositionindex |
| Clusters 1 and 3 | Clusters 4 and 5 | |
| Cluster 1 | Cluster 3 | timesinceally, independence |
| Cluster 4 | Cluster 5 | |
| Cluster 4 | Cluster 5 | treaties, conferences, weightedunvote, unweightedunvote, intergovorgs, ngo, officialvisits, exportbooks, relexportbooks, tourism, reltourism, tourism3, exports, militaryalliance, commonbloc2 |

Table S3: Top blocks from the *Nations* data analysis. The countries clusters are described in Supplementary Section D.

## C  Time complexity

The total cost of our Algorithm 1 is $\mathcal{O}(d)$ per iteration, where $d = \prod_k d_k$ denotes the total number of tensor entries. The per-iteration computational cost scales linearly with the sample size, and this complexity is comparable to the classical tensor methods such as CP and Tucker decomposition. More specifically, each iteration of Algorithm 1 consists of updating the core tensor $\mathcal{C}$ and $K$ membership matrices $M_k$'s. The update of $\mathcal{C}$ requires $\mathcal{O}(d)$ operations and the update of $M_k$ requires $\mathcal{O}(R_k \frac{d}{d_k})$ operations. Therefore the total cost is $\mathcal{O}(d + d \sum_k \frac{R_k}{d_k})$.

## D  Additional information for real data analysis

**Multi-tissue gene expression.** The gene expression data we analyzed is part of the GTEx v6 datasets (`https://www.gtexportal.org/home/datasets`). We cleaned and preprocessed the data following the steps in [2]. We focused on the 13 brain tissues, 193 individuals, and 362 annotated genes provided by Atlax of the Developing Human Brain (`http://www.brainspan.org/ish`). After applying the $\ell$-0 penalized TBM to the mean-centered data tensor, we identified the following four clusters of tissues:

- Cluster 1: Substantia nigra, Spinal cord (cervical c-1)
- Cluster 2: Cerebellum, Cerebellar Hemisphere
- Cluster 3: Caudate (basal ganglia), Nucleus accumbens (basal ganglia), Putamen (basal ganglia)
- Cluster 4: Cortex, Hippocampus, Anterior cingulate cortex (BA24), Frontal Cortex (BA9), Hypothalamus, Amygdala

We found that most tissue clusters are spatially restricted to specific brain regions, such as the two cerebellum tissues (cluster 2), three basal ganglia tissues (cluster 3), and the cortex tissues (cluster 4). Supplementary Table S2 reports the associated gene cluster for each tissue cluster. Because our method attaches importance to blocks by the absolute mean estimates, our method is able to detect both over- and under-expression patterns. Blocks with highly positive means correspond to over-expressed genes, whereas blocks with highly negative means correspond to under-expressed genes.

**Nations dataset.** This is a $14 \times 14 \times 56$ binary tensor consisting of 56 political relations of 14 countries between 1950 and 1965 [3]. The tensor entry indicates the presence or absence of a political action, such as "treaties", "sends tourists to", between the nations. We applied the $\ell$-0 penalized TBM to the binary-valued data tensor, and we identified the following five clusters of countries:

- Cluster 1: Brazil, Egypt, India, Israel, Netherlands
- Cluster 2: Burma, Indonesia, Jordan
- Cluster 3: China, Cuba, Poland, USSA
- Cluster 4: USA
- Cluster 5: UK

Supplementary Table S3 reports the cluster constitutions for top blocks. Because the tensor entries take value on either 0 or 1, the top blocks mostly have mean one.