[Reviews · NeurIPS 2019]

Reviewer 1



Comment on rebuttal: Thanks for addressing my concerns. I increased my score. By my comment on \lambda on table 1, I meant that the bottom of the table hides the bar not that the definition of \bar{\lambda} is not clear. --------------------------------------------------------------------------------------------------------- The paper studies the problem of recovering block structure from a noisy tensor, which can be seen as a higher-order extension of stochastic block models. A least square estimator with good convergence properties is proposed together with alternating least squares implementation. Extensions are proposed to select number of blocks as well as model sparse data. This is a well-executed paper. The formulation is clear and the flow of ideas is natural. Theoretical analysis and comprehensive experiments are provided. I did not check the correctness of the proofs. I do have some minor comments though: - L59: I believe by "fiber" the authors mean "slice". - L131: How do you know your estimator is "nearly optimal"? If this is based on equation (6) I would remove it and maybe mention later than equation (6) provides a *suggestive evidence* for optimality. - L230: Please indicate that this is *RMSE* rate and use big O notation to make it clear you dropped a term. - In sparsity experiment, it seems you are using \rho instead of p to indicate sparsity? Please fix that and specify the value of \rho (C norm) you used. - Also, in sparsity experiment. It does not make much sense to report baseline results. They do not add any information and only make the table more difficult to read. - In the description of table 1: the bar of \lambda is not clear.

Reviewer 2



I have read the author response and found that the authors significantly addressed my concerns -- they provided new theoretical and empirical results. The results are convincing, and I increase my score. ------- In this paper, a tensor block model -- a multiway extension of a stochastic block model -- is studied. The main contribution is to derive a statistical convergence rate of the least square estimator under sub-Gaussian noise. The authors try to confirm the theoretical results by numerical simulations. The strength of this paper is in the theoretical result, which improves the existing convergence rate and also proves the consistency of clustering results. However, my current evaluation is slightly below the border of acceptance. The follows are my major concerns. Originality. The problem of tensor block models has been at least studied since Jegelka et al. (2009), who proposed an efficient algorithm with an approximation guarantee. Chi et al. (2018) derived a statistical convergence rate earlier than this work. Thus, strictly saying, the originality of this paper is to improve the convergence rate. Although a few extensions such as sparse estimation is proposed, I feel they are somewhat incremental. Measuring MSE in a clustering problem. The convergence rate studied in this paper is for the mean square error between a true (noiseless) tensor and a recovered tensor. I agree with this setting if it is for a tensor recovery problem. However, how about the MSE setting for a clustering problem? For example, suppose we have two estimators A and B such that MSE(A) <= MSE(B). Can we say the clustering result of A is always better than B? I mean, it seems there is a gap between MSE and the correctness of the clustering, and I'm not sure measuring MSE is a right thing for clustering. Toy data experiments. In Figure 2, I'm not convinced that the results are consistent with the theory. Specifically, I feel a gap between (4,4,4) and others. This may be because the range of the x axis is different between (4,4,4) and others after rescaled N. Also the number of R settings is not large enough. Real data experiments. The baselines, CP and Tucker decompositions, are general methods for tensor decomposition but not proper methods for clustering. It should be compared with Jegelka et al. (2009) and Chi et al. (2018).

Reviewer 3



Thank you for addressing my comments in the rebuttal. I increased my score. ----------------------------------------------------- This paper proposes a tensor block model clustering method with applications to multiview clustering. They propose an optimization method for the clustering model using the least-squares method. The proposed method is supported with theoretical analysis on convergence. Furthermore, the paper is well written with supporting experiments. I am not an expert on this topic, however, my judgment is that the paper borrows similar ideas from [15] and other papers and give an extension. Still, I feel the paper has certain among of novelty in the proposed clustering method and shows good performance. I feel that the paper makes a reasonable contribution. Some issues in the supplementary section: What do the authors mean by A.2 in the second proof on page 1 of the appendix? Also, I did not understand this "Combining (A.2), (A.2) and (A.2), we have". In fact, some of the equation references in the appendix are confusing.

[Author Response · NeurIPS 2019]

We thank the reviewers for the helpful comments and feedback. Our responses are detailed below.

**To Reviewer 1.**

- In the description of Table 1: $\bar{\lambda}$ is the sparsity penalty parameter averaged across 50 simulations.

- Wording in L59, L131, L230. We will make the suggested edits for clarity.

- Sparsity experiment and Table 1. We will remove the baseline case ($\lambda = 0$) from Table 1. The "sparsity $\rho$" in the description should be corrected to "sparsity ($p$)".

- Real data: We will add the following additional analysis to the Section *7.3. real data*. Specifically, we ran the clustering analysis on the *Brain expression* and *Nations* datasets and then compared the goodness-of-fit of different methods. Because the code of CoCo method [Chi et al, 2018] is not yet available as of 07/31/2019, we excluded it from our numerical comparison (we did have a theoretical comparison with CoCo). The following table summarizes the proportion of variance explained by each clustering method:

Table: Comparison of goodness-of-fit in the *Brain* expression and *Nations* datasets.

| Dataset | TBM | TBM-sparse | CP | Tucker | CoTeC [Jegelka et al 2009] | CoCo [Chi et al 2018] |
|---|---|---|---|---|---|---|
| Brain expression | 0.856 | 0.855 | 0.576 | 0.434 | 0.849 | - |
| Nations | 0.439 | 0.433 | 0.324 | 0.253 | 0.419 | - |

Our method (TBM) achieves the highest variance proportion, suggesting that the entries within the same cluster are close (i.e., a good clustering). As expected, the sparse TBM results in a slightly lower proportion, because it has a lower model complexity at the cost of small bias. It is remarkable that the sparse TBM still achieves a higher goodness-of-fit than others. The improved interpretability with little loss of accuracy makes the sparse TBM appealing in applications.

**To Reviewer 2.**

- Measuring MSE in a clustering problem. We agree with reviewer that MSE is not the best metric for clustering. In fact, Theorem 2 of our paper provides a consistency result for mis-classification rate (MCR) specifically for clustering. In addition, we also compared the empirical clustering error rate (CER, i.e., 1 - rank index) in the simulation. Both metrics, combined with the MSE, provided a fair comparison in the clustering problem. Following the reviewer's suggestion, we now upgrade the consistency result to a finite-sample convergence rate and will add the result below to the final paper.

**Theorem 0.1** (Simplified version). *Consider a Gaussian tensor block model with variance parameter $\sigma^2$ and non-degenerate clusterings. In the case when $d_1 = \ldots = d_K = d$ and $R_1 = \ldots = R_K = R$, we have*

$$\mathbb{P}(MCR(\hat{\boldsymbol{M}}_k, \boldsymbol{P}_k \boldsymbol{M}_{k,true}) \geq \varepsilon) \leq 2R^{K(d+1)} \exp\left(-\frac{C_2 \delta_{\min}^2 d^K \varepsilon^2}{\sigma^2 R^{2K}}\right), \quad \text{for all } k \in [K],$$

*where $\boldsymbol{P}_k \in \mathbb{R}^{R_k \times R_k}$ is a permutation matrix, $C_2 > 0$ is a constant independent of tensor dimension, and $\delta_{\min} > 0$ is the minimal gap (under some natural measurement) between tensor block means.*

The above result implies that the clustering error converges to zero at the rate of $\mathcal{O}\left(\frac{\sigma \sqrt{\log R}}{d^{(K-1)/2} \delta_{\min}}\right)$. Here the block-mean gap $\delta_{\min}$ serves the role of the eigen-separation as in classical tensor Tucker decomposition.

- Data experiments. Regarding the real data, please see our response to Reviewer 1. Regarding the toy data, we have added the suggested numbers of clusters. See the figure below. The added curves now fill in the gap in the figure.

**To Reviewer 3.**

- Novelty. Our method is related to, but also clearly distinctive from, previous methods in three aspects: accuracy, interpretability, and scalability. The table below summarizes the comparison. In practice, our TBM method performs favorably in both simulation and real data (see our newly added analysis in response to Reviewer 1). Since tensor-valued data is now common in a number of fields, we believe this work will be of interest to the community.

| Method | Recovery error (MSE) | Clustering error (MCR) | Block detection | Time complexity (flop / iter) |
|---|---|---|---|---|
| Tucker (rigorous; $K=3$) | $dR$ | $\frac{\sigma\sqrt{R}}{d\lambda_{\min}}$ up to rotation | No | $d^K$ |
| CoCo [Chi et al 2018] (rigorous) | $d^{K-1}$ | - | No | $d^{K+1}$ or $d^K$ |
| TBM (rigorous, this paper) | $d\log R$ | $\frac{\sigma\sqrt{\log R}}{d^{(K-1)/2}\delta_{\min}}$ | Yes | $d^K$ |
| Optimal rate [Gao et al 2018] (heuristic) | $d\log R$ | - | - | - |

– Regarding the equation numbers in the Supplement, we will correct them in the final version.

[Meta-Review · NeurIPS 2019]

After reading the rebuttal, the reviewers agreed that most of their concerns had been addressed and that this paper proposes an interesting idea with potential applications that will be of interest to the community.